# Piezo1 regulates the mechanotransduction of soft matrix viscoelasticity

**Mariana A. G. Oliva** [1,2], **Giuseppe Ciccone** [1,2], **Gotthold Fläschner**[2], **Jiajun Luo**[1], **Jonah L. Voigt** [1,3,4], **Patrizia Romani** [5], **Paul Genever**[6], **Oana Dobre** [1], **Sirio Dupont** [5], **Massimo Vassalli** [1], **Pere Roca-Cusachs** [2,7] ✉ & **Manuel Salmeron-Sanchez** [1,2,8] ✉

Mechanosensitive ion channels such as Piezo1 have fundamental roles in sensing the mechanical properties of the extracellular matrix. However, whether and how Piezo1 senses time-dependent matrix mechanical properties, that is, viscoelasticity, remains unknown. To address this question, we combine an immortalised mesenchymal stem cell line, in which Piezo1 expression can be silenced, with soft and stiff viscoelastic hydrogels that have independently tuneable elastic and viscous moduli. We demonstrate that Piezo1 is a regulator of the mechanotransduction of viscoelasticity in soft matrices, both experimentally and through simulations incorporating Piezo1 into a modified viscoelastic molecular clutch model. Using RNA sequencing, we also identify the transcriptomic responses of mesenchymal stem cells to matrix viscoelasticity and Piezo1 activity, identifying gene signatures that reflect their mechanobiology in soft and stiff viscoelastic hydrogels.

It is now widely accepted that the mechanical properties of the extracellular matrix (ECM) drive cellular behaviour, such as differentiation, proliferation and migration[1–3]. The dynamic molecular interactions between cells and their surrounding matrix have been explained by the existence of a molecular clutch formed by the cell's actin cytoskeleton, myosin contractile motors and the adaptor proteins that connect ECM binding integrins to the actin cytoskeleton. The molecular clutch model was first introduced by Chan and Odde[4] to mechanistically describe how cells sense and respond to the mechanical properties of the ECM. As a result of this model, cell response to matrix stiffness (also referred to as rigidity)[5,6], viscosity[7] and viscoelasticity[8,9] has been described via these mechanisms in more evolved versions of this framework.

In light of the molecular clutch model, studies of the cell response to different matrix mechanics have focused on mechanisms of cell-ECM adhesion, as well as on adaptor proteins and ECM molecules[10]. Beyond these elements, transmembrane ion channels such as Piezo1 could also play an important role, given their known mechanosensing properties[11]. In 2010, the Piezo1 channel was identified in a high-throughput screen for integrin co-activators in epithelial cells and was later categorised as being a mechanically activated cation channel by Ardem Patapoutian and colleagues[12,13]. Recently, a relationship between Piezo1 and integrin-mediated focal adhesion (FA) signalling has been described[14–17], in which Piezo1 channel activity is proposed to act as a key mediator of integrin signalling, and therefore of cell-ECM interactions.

Previous studies have found that substrate stiffness alone is sufficient to enhance Piezo1-mediated $Ca^{2+}$ signalling, which in turn influences the differentiation of neural stem cells[18]. Substrate elasticity has been shown to drive cell function via the engagement of the actin-talin-integrin-fibronectin (FN) molecular clutch[6], indicating that mechanotransduction depends on a stiffness threshold that promotes clutch engagement. Because of this, and on account of the coordinated action of Piezo1 activity and integrin signalling[17], we hypothesise that Piezo1 expression affects the engagement of the molecular clutch

[1]Centre for the Cellular Microenvironment (CeMi), The Advanced Research Centre, University of Glasgow, Glasgow, UK. [2]Institute for Bioengineering of Catalonia (IBEC), The Barcelona Institute for Science and Technology (BIST), Barcelona, Spain. [3]Max Planck Institute for Medical Research, Heidelberg, Germany. [4]Cellular Biomechanics, Faculty of Engineering, Bayreuth University, Bayreuth, Germany. [5]Department of Molecular Medicine (DMM), University of Padua, Padua, Italy. [6]Department of Biology, University of York, York, UK. [7]University of Barcelona, Barcelona, Spain. [8]Institució Catalana de Recerca i Estudis Avançats (ICREA), Barcelona, Spain. ✉e-mail: proca@ibecbarcelona.eu; msalmeron@ibecbarcelona.eu

and downstream mechanotransduction in mesenchymal stem cells (MSCs).

However, native ECMs do not behave as perfectly elastic solids. Instead, in assessment of their response to mechanical deformation, reconstituted ECMs have been shown to initially resist deformation followed by a time-dependent dissipation of energy, a property that is characteristic of viscoelastic materials[8]. Energy dissipation arises from the dynamic molecular organisation of the ECM, which is not a perfectly chemically crosslinked network, but instead features the breaking of weak bonds[19], protein unfolding[20] and entanglements release[8,21]. The interaction of cells with a viscoelastic substrate has been modelled via the same molecular clutch mechanism[9,22]. However, how and whether Piezo1 senses ECM viscoelasticity within this framework remains unknown.

In this study, we used two pairs of viscoelastic polyacrylamide (PAAm) hydrogels, with either a low or high elastic component (Young's modulus $E = 0.4$ kPa and 25 kPa), each with a higher or lower dissipative component, respectively. By combining these hydrogels with a mechanosensitive immortalised (Y201) MSC line in which we could transiently knock down Piezo1 expression, we demonstrated that Piezo1 mediates viscoelasticity sensing within the molecular clutch framework, especially at low substrate stiffness. Specifically, we found that enhanced energy dissipation in low-stiffness hydrogels promotes cell spreading, FA formation and overall molecular clutch engagement in a Piezo1-dependent manner. Critically, the molecular clutch model was extended in this work to account both for the cell interactions with dissipative substrates at different stiffness ranges as well as for the overall effect of Piezo1 in clutch engagement. Simulations with the extended model captured our observed experimental behaviours of cells on viscoelastic matrices. Furthermore, these results were consistent with downstream mechanotransduction events, including enhanced cell metabolic capacity and transcriptional adaptation. Using RNA sequencing (RNAseq), we identified differentially regulated genes that mediate the cells' response to substrates stiffness, viscoelasticity and Piezo1 expression. Our results therefore indicate that Piezo1 is a mechanotransducer of time-dependent ECM mechanics in addition to substrate stiffness.

## Results

### Cell response to matrix viscoelasticity is stiffness- and Piezo1-dependent

Two different hydrogel pairs were synthesised by mixing different amounts of acrylamide (AAm) and bis-acrylamide (BisAam) to obtain hydrogels of approximately the same $E$ but with varying stress relaxation rates. To achieve soft ($E \approx 0.4$ kPa) and stiff ($E \approx 25$ kPa) hydrogels, we combined two previously reported strategies to tune viscoelasticity in PAAm hydrogels (Fig. 1a).

To generate the stiff hydrogel pair, we used the method first reported by Cameron and colleagues[23,24] and later optimised by our group[25], in which substrate viscoelasticity is mediated by the movement of loosely crosslinked polymer chains. To create a softer pair of viscoelastic gels, we used the approach reported by Charrier and colleagues[26,27], as the Cameron et al. approach, when modified to create softer hydrogels, gave rise to sticky hydrogels that were difficult to handle. In the Charrier et al. method, substrate viscoelasticity arises from physically entrapped chains of high molecular weight, linear Aam (Fig. 1a). The resulting hydrogel groups had a Young's modulus of approximately 0.4 kPa and 25 kPa (hereafter referred to as *soft* and *stiff*, respectively); we observed no significant differences in Young's modulus within each stiffness group (Fig. 1b). To characterise the differences in the hydrogel's stress relaxation behaviour, we performed stress relaxation measurements with a physiologically relevant step strain ($\varepsilon$) of 7%[28,29] applied over 60 s using nanoindentation. From the resulting stress relaxation curves (Fig. 1c, d), we calculated the time for the stress to relax to 80% of the initial value (Supplementary Data Fig. 1a), as well as

the % of energy dissipation for each hydrogel condition (Fig. 1e). Resulting data demonstrated that for each stiffness group, there was a slow-relaxing (V-, elastic) and fast-relaxing (V+, viscoelastic) hydrogel, and that the V+ hydrogels relax ~2 times faster than their elastic counterparts (Fig. 1c, d), and display higher relaxation amplitude, shown as % of energy dissipated (Fig. 1e). To confirm data obtained by nanoindentation, we additionally performed shear bulk rheology measurements. By computing the ratio between the loss modulus ($G''$) and the storage modulus ($G'$), we observed that the *tan (δ)* of the V+ hydrogels increased for both stiffness groups with respect to V- hydrogels (Supplementary Data Fig. 1b, c), emulating soft tissues that exhibit loss moduli of approximately 10% of their elastic moduli at 1Hz[8]. Notably, as PAAm hydrogels are chemically crosslinked, both strategies give rise to viscoelastic solids with no plastic deformation, in contrast to physically crosslinked viscoelastic hydrogels[30]. These properties allowed us to investigate the cell response to substrate viscoelasticity independently of substrate elasticity, or Young's modulus, in two distinct stiffness regimes in 2D. Moreover, we performed nanoindentation curve contact point analysis to rule out any potential differences in hydrogels' topography due to i) the presence of linear acrylamide in the soft group and ii) the different ratios of monomer and crosslinker in the stiff group (Supplementary Data Fig. 1f, g). Next, we checked that the ECM matrix protein FN, the ECM element of the molecular clutch, was homogeneous on all substrates regardless of their stiffness and viscoelasticity, as previously reported[23,26]. By using the crosslinker sulfo-SANPAH, FN was conjugated on each hydrogel[31]. By using immunofluorescence, we confirmed a homogenous FN coating on all substrates with no significant changes in signal intensity (Supplementary Data Fig. 1d, e). Overall, these findings show that we have established a hydrogel system that spans a wide range of physiologically relevant stiffnesses with different rates of stress relaxation, allowing for the investigation of the effects of ECM viscoelasticity on cell behaviour independently of ligand density and topography.

We next investigated Piezo1 gene expression of control Y201 MSCs (scRNA) and of MSCs in which Piezo1 was knocked down via small interfering RNA (siRNA) for Piezo1 using an siRNA screen (siRNAs 1–3) (Supplementary Data, Fig. 2a). Additionally, we assessed morphology and adhesions on all screened conditions plus on cells treated with the mechanosensitive ion channel inhibitor GsMTx4 (Supplementary Data, Fig. 2b, c). This allowed us to select the most consistent siRNA, siRNA1, which was used subsequently in all experiments and referred to as siPiezo1. siPiezo1-mediated Piezo1 knock down demonstrated clear phenotypic differences in cell morphology and adhesion formation (Supplementary Data, Fig. 2d), as well as approximately 50% efficiency in Y201 MSCs in terms of protein and gene expression levels (Supplementary Data, Fig. 2e, f). Following this, we assessed the morphology of scRNA and siPiezo1 MSCs on all hydrogel conditions (representative images in Fig. 1f). On soft hydrogels, scRNA MSCs increased their spreading area in response to substrate stress relaxation, whereas on stiff substrates, the opposite occurred (Fig. 1g, h). This response was also reflected in terms of their circularity (Fig. 1i, j). These results corroborate data first reported by Chaudhuri and colleagues in cancer epithelial cells[9], which indicated that cell spreading in response to increased substrate relaxation is stiffness-dependent and only increases at lower stiffness values in stress-relaxing substrates (<1pN/nm, or approximately 1 kPa[22]).

In terms of Piezo1-dependent morphology, knock down has previously been reported to disrupt actin fibres and integrin activity, thus decreasing cell spreading[32]. In this study, siPiezo1 MSCs decreased their spreading area and increased their circularity, as compared to scRNA MSCs, when cultured on soft viscoelastic (V+) or stiff elastic (V-) hydrogels (Fig. 1k, l). Interestingly, the cell-spreading response to viscoelasticity seen in scRNA MSCs was abrogated in siPiezo1 MSCs cultured on low stiffness (soft, 0.4 kPa) but not on stiff substrates (stiff, 25 kPa), as these cells had presumably reached their minimum

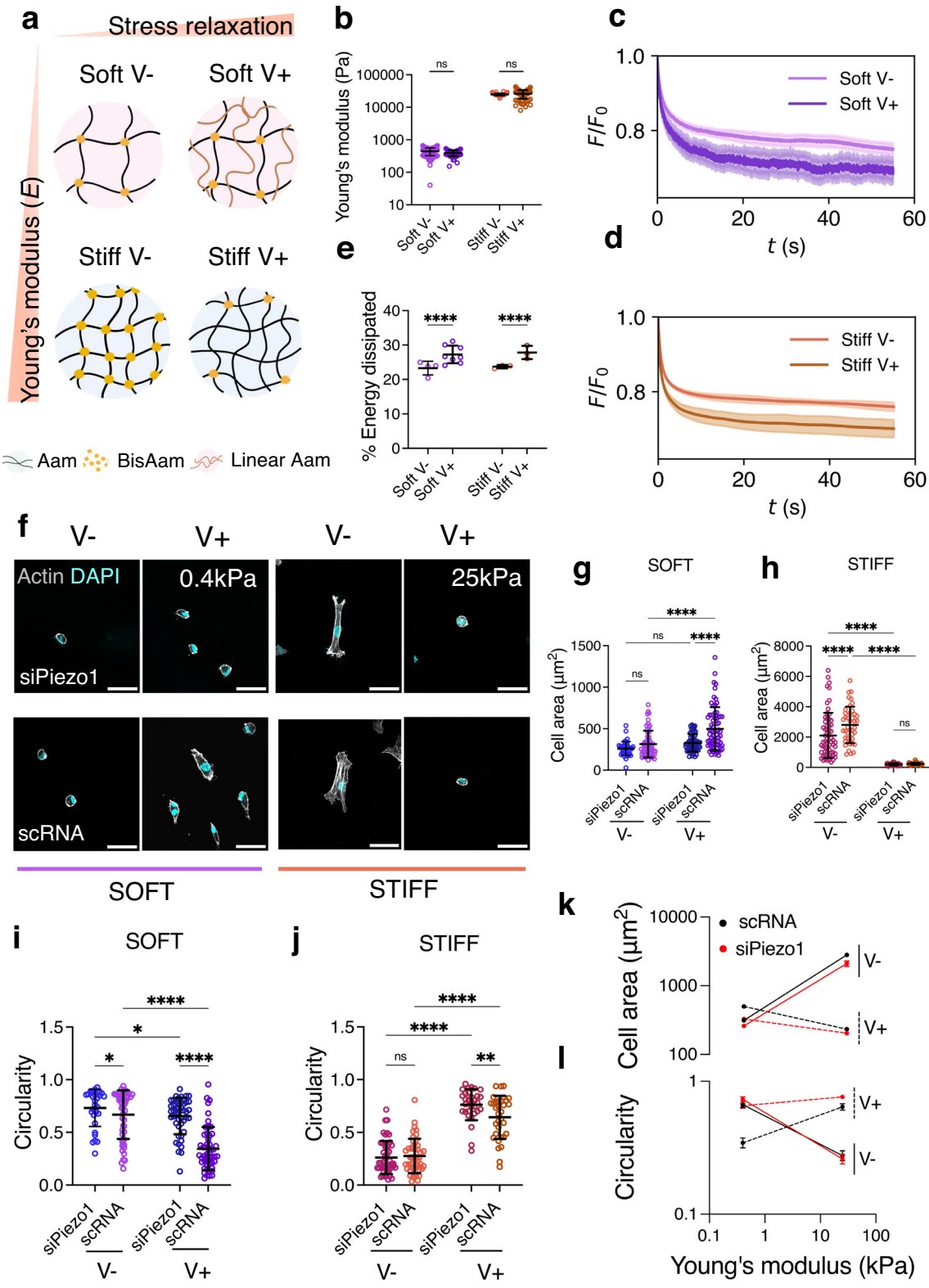

spreading area independently of Piezo1. This Piezo1-mediated cell spreading behaviour was also observed when we cultured MSCs on FN-coated glass substrates (Supplementary Data, Fig. 2d, e, f), where siPiezo1 cells showed reduced cell spreading area compared to scRNA MSCs. These data endorse the overall suppressive effect of channel knock down on cell-substrate interaction, as previously reported in other cell types[13]. Together, our results identify a Piezo1-dependent response to matrix viscoelasticity in soft matrices.

**Molecular clutch engagement in response to matrix viscoelasticity is Piezo1-dependent**

To further investigate the mechanism of Piezo1-mediated ECM viscoelasticity sensing, we quantified the engagement of the actin-talin-integrin-FN molecular clutch in MSCs cultured on the different substrates in the presence and knock-down of Piezo1. Clutch dynamics regulate cell mechanotransduction in response to substrate stiffness[6], viscosity[7] and more recently, viscoelasticity[9,22]. However, how and

**Fig. 1 | Cell response to matrix viscoelasticity is stiffness and Piezo1 dependent. a** Representation of the different hydrogel networks. Here and in the other figure panels, V− represents an elastic, slow-relaxing hydrogel and V+, a fast-relaxing, viscoelastic hydrogel. Aam: Acrylamide, BisAam: Bisacrylamide, Linear Aam: Linear Acrylamide. Gel network diagram was created with BioRender **b** Nanoindentation measurements, showing Young's modulus data obtained for Soft (0.4 kPa, purple) and Stiff (25 kPa, orange) hydrogel pairs. Data shown as individual values, mean ± SD (from left to right $n = 95, 58, 113, 80$ single indentation curves, from $N \geq 3$ gels). **c** Representative average stress relaxation curve ± SD of an indentation performed on soft (purple) and **d** stiff (orange) hydrogels. **e** Average energy dissipated in soft (purple) and stiff (orange) hydrogel groups, shown as individual values, mean ± SD. From left to right $n = 4, 8, 3$ and $3$ indentations maps, comprising 61, 87, 41 and 71 total indentations on $N = 3$ hydrogels. **f** Representative Actin and DAPI immuno-fluorescence images of control (scRNA) and Piezo1 knock down (siPiezo1) Y201 MSCs cultured on different hydrogel groups for 48 h. Scale bar = 50 μm

**g, h** Quantified cellular area on the soft (left) and stiff (right) hydrogel groups. All individual points represent individual cell measurements. Data shown as individual values, mean ± SD, in (**g**) from left to right $n = 47, 58, 59$ and $66$ cells; in (**h**) from left to right $n = 62, 42, 40$ and $33$ cells. Data from $N = 3$ independent experiments. **i, j** Quantified cellular circularity of the soft (left) and stiff (right) hydrogel groups. Data shown as individual values, mean ± SD. In (**i**) from left to right $n = 32, 50, 47$ and $51$ cells; in (**j**) from left to right n = b 52, 46, 31 and 33 cells. Data from $N = 3$ independent experiments. **k** Summary of mean cellular area ± SEM plotted as a function of stiffness for scRNA (black), siPiezo1 (red), V− (continuous line) and V+ (dashed line) conditions. **l** Summary of mean cellular circularity ± SEM plotted as a function of stiffness for for scRNA (black), siPiezo1 (red), V− (continuous line) and V+ (dashed line) conditions. Statistical analyses were performed using a two-way ANOVA test. *P* values indicating significance, ns > 0.05, *≤0.05, **≤0.01, ***≤0.001, ****≤0.000. Specific *p* values and descriptive statistics are provided in the Source Data.

whether Piezo1 relays ECM viscoelasticity cues via the molecular clutch is unknown. We first quantified FA formation by looking at vinculin using immunofluorescence, as vinculin is recruited to adhesion sites in response to sustained force-sensing by the actin-talin-integrin-FN clutch[6] (Fig. 2a). Individual vinculin FA length (Fig. 2b, c) and FA count (Supplementary Data Fig. 3a–c) were quantified in siPiezo1 and scRNA MSCs cultured on the soft and stiff hydrogel groups. The resulting data supported the previously observed cell-spreading phenotype: FA length increased in scRNA MSCs under conditions of faster substrate stress relaxation in a soft regime (0.4 kPa), while FA length decreased in scRNA MSCs cultured on stiff (25 kPa) hydrogels in response to enhanced stress relaxation (Fig. 2d).

Since Piezo1 was first identified in 2010 as an integrin co-activator, its activity has been intrinsically linked to FA dynamics[13–17,32]. Indeed, Piezo1 knock down was sufficient to visibly reduce FA size and number in Y201 MSCs cultured on FN-coated glass substrates (Supplementary Data Fig. 2e). Unlike scRNA MSCs, siPiezo1 MSCs did not exhibit increased FA size (Fig. 2b) or number (Supplementary Data Fig. 3a) when seeded on soft, fast-relaxing matrices (soft V+) as compared to elastic (soft V−) substrates. This highlights the fundamental role of this channel in mediating cell response to matrix viscoelasticity at low substrate stiffness. Both siPiezo1 and scRNA MSCs exhibited significantly decreased FA length (Fig. 2c) and number (Supplementary Data Fig. 3b) when cultured on stiff V+ substrates, compared to stiff V-. However, the magnitude of this change was lessened in siPiezo1 MSCs. Similarly, when culturing scRNA and siPiezo1 MSCs on FN coated glass substrates, siPiezo1 MSCs had fewer FA structures compared to scRNA MSCs (Supplementary Data Fig. 2f). These results further established that Piezo1 channel activity is inherently linked to FA dynamics.

We then tested whether a computational model of the molecular clutch could capture both the effect of stiffness in mediating cell response to viscoelasticity, as previously reported[9,22] and as seen experimentally in this work; and the Piezo1-dependent clutch engagement on soft matrices. The model, based on our previous work[6], simulates the stochastic cell adhesion dynamics on a viscoelastic substrate using a Monte Carlo approach that captures integrin-ligand binding, force transmission and adhesion growth (Fig. 2e). To model the response to a viscoelastic substrate, we modified our previous model to consider a viscoelastic rather than elastic substrate, modelled as a Standard Linear Solid (SLS). The SLS exhibits instantaneous resistance to deformation and time-dependent relaxation, as observed experimentally (see Fig. 1c, d), for details see Supplementary Note 1. Upon integrin binding to the substrate-bound ligands, myosin-generated forces build up in the clutch, leading to talin unfolding, vinculin binding and subsequent adhesion growth (modelled as integrin recruitment). Taking the soft elastic condition as reference, increasing stiffness (soft V− vs stiff V− conditions) results in higher forces and hence greater FAs in experiment (Fig. 2d) and theory

(Fig. 2h). Increasing viscosity (soft V- versus soft V+ conditions) also leads to greater adhesion growth in experiment (Fig. 2b) and theory (Fig. 2f). This can be explained from both additional forces due to viscous resistance, and the effect of time-varying elasticity and hence force, affecting clutch buildup and lifetime[22]. However, within the viscoelastic conditions, increasing stiffness (soft V+ versus stiff V+ conditions) reduces adhesion length (Fig. 2d), and thus integrin recruitment. To explain this in the model, this requires considering that integrin recruitment does not occur instantaneously upon talin unfolding, as assumed for simplicity in our previous models on elastic substrates. Instead, here we explicitly consider the timescale of this process, captured by the model via a stochastic, binding rate of vinculin to unfolded talin. In this scenario, clutch dynamics in the stiff condition become faster than vinculin binding, decreasing integrin recruitment (Fig. 2g). In summary, the model closely recapitulates the experimental effects of stiffness and viscosity.

Regarding the effects of Piezo1, different works have shown that it is involved in integrin activation and integrin adhesion maturation[14]. We thus modelled it by increasing integrin unbinding rates ($k_{off}$) by 15% in siPiezo1 MSCs. With this, we reproduced the experimental effects of siPiezo1, which reduced integrin recruitment induced by both increased stiffness (soft V− versus stiff V−) and increased viscosity (soft V− versus soft V+). In the simulations, the viscous effect for low stiffness substrates is not entirely abolished. Whereas a more pronounced reduction of $k_{off}$ would allow for closer matching the experimental data on integrin densities, the results for the retrograde flow (Fig. 3) would not be consistent with experimental results. Overall, the modified clutch model (Fig. 2g) closely matches our experimental observations (Fig. 2d), which (i) describe a stiffness-dependent response to clutch engagement in viscoelastic matrices, (ii) highlight how Piezo1 regulates clutch activation and (iii) propose Piezo1 a key mediator of cell response to soft matrix viscoelasticity.

FA maturation and size are inversely related to the actin retrograde flow rate in cells[33], because when the molecular clutch is engaged via actin-talin-integrin-FN links, actin is bound in its fibrillar form and the rate of actin polymerisation decreases. We therefore transfected siPiezo1 and scRNA MSCs with Life-Act GFP, which stains filamentous actin (F-actin) structures (Fig. 3a), and measured the rate of actin retrograde flow using live confocal microscopy, in response to varying viscoelasticity on soft and stiff substrates. In accordance with FA data (Fig. 2b–d), actin retrograde flow was slowed in scRNA MSCs in response to faster substrate stress relaxation in soft hydrogels (Fig. 3b), whereas on stiff hydrogels, actin retrograde flow increased ~7 fold from a slow-relaxing (stiff V−) to a fast-relaxing (stiff V+) matrix in scRNA MSCs and ~3 fold in siPiezo1 MSCs (Fig. 3c). Notably, in siPiezo1 MSCs, the knocking down of Piezo1 abrogated any change in retrograde flow speed between V− and V+ conditions in the soft group (Fig. 3b). These data (Fig. 3b–d) reiterate the role of Piezo1 in finely sensing the time-dependent mechanics of soft matrices and in

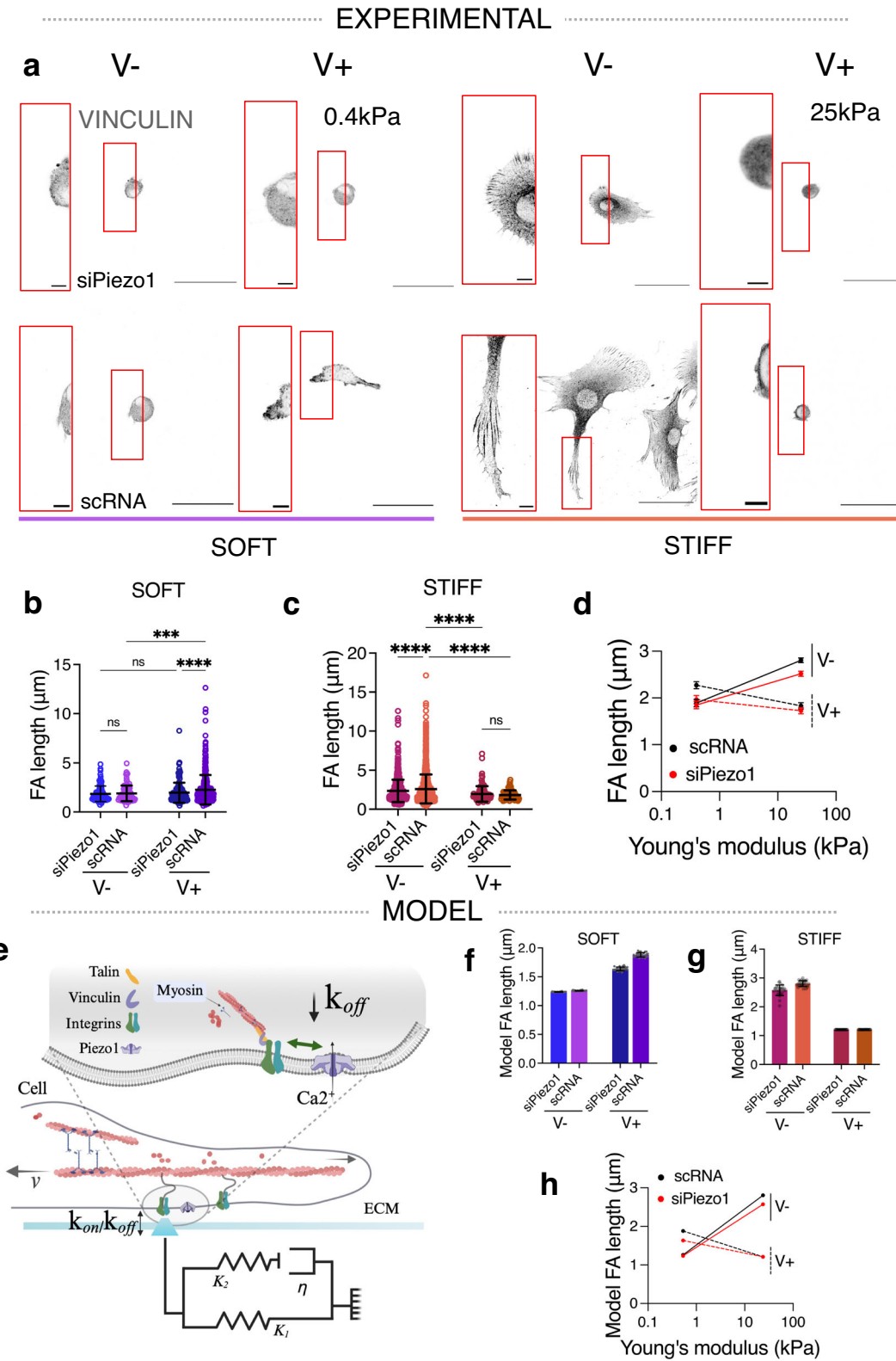

promoting clutch engagement in response to enhanced viscoelasticity on soft matrices.

Using the same modified clutch model as before (Fig. 2e), we assessed how stress relaxation and Piezo1 activity influence the retrograde actin flow speed of cells on soft (Fig. 3e) and stiff (Fig. 3f) hydrogels. Again, the modified clutch model (Fig. 3g) closely recapitulated our experimental observations (Fig. 3d), whereby on soft matrices, scRNA cells decrease actin flow rate in response to enhanced

stress relaxation whereas on stiff substrates, scRNA MSC actin retrograde flow speed is increased. Moreover, on soft matrices, siPiezo1 cells did not reduce actin retrograde flow speed in response to enhanced energy dissipation, which proposes Piezo1 as a key regulator of stress relaxation-generated clutch activation at this stiffness.

Finally, to investigate how the observed adhesion phenotype and actin polymerisation might translate into cell-exerted forces transmitted to the underlying ECM, we quantified the average cell traction

**Fig. 2 | Focal adhesion formation in response to matrix viscoelasticity is stiffness and Piezo1-dependent. a** Representative images of immunostained vinculin adhesions in siPiezo1 and scRNA Y201 MSCs cultured on soft (0.4 kPa, left) and stiff (25 kPa, right) hydrogel groups of varying stress relaxation rates. Scale bar = 50 μm, zoomed image scale bar = 5 μm. **b, c** Quantified individual focal adhesion (FA) length of cells on the soft (left) and stiff (right) hydrogel groups. In (**b**) from left to right $n = 103$, 104, 130 and 371 individual FA measurements from 30, 31, 32 and 39 cells; in (**c**) from left to right $n = 689$, 1289, 103 and 76 individual FA measurements from 30, 31, 31 and 35 cells. Data from $N = 3$ independent experiments. Data shown as individual values, mean ± SD. **d** Summary of mean individual FA length ± SEM plotted as a function of stiffness for scRNA (black), siPiezo1 (red), V− (continuous line) and V+ (dashed line) conditions. Statistical analyses were performed using a two-way ANOVA test. *P* values indicating significance, ns > 0.05, *≤0.05, ***≤0.001, ****≤0.0001. Specific *p* values and descriptive statistics are provided in the Source Data. **e** Schematic of the influence of

Piezo1 in molecular clutch engagement. Cell is shown to be coupled to the ECM via ECM-binding integrins that in turn, connect to the contractile actin filaments via mechanosensitive adaptive proteins (talin and vinculin). Myosin motors continuously pull on actin filaments with velocity (*v*). Here, the ECM is modelled as a Standard Linear Solid (SLS), composed of two elastic springs ($K_1$ and $K_2$) and a viscous dashpot element ($\eta$). $k_{on}$ and $k_{off}$ represent the rates of clutch association and dissociation, respectively. In the zoom (grey shading), the concerted action between Piezo1 and integrins is highlighted, showing that potentiation of this interaction decreases clutch dissociation ($k_{off}$), created with BioRender. **f, g** Scaled model predictions of focal adhesion length (μm) on the soft (left) and stiff (right) substrate groups. Data shown as individual values, mean ± SD ($N = 26$ simulations). **h** Summary of model predictions for mean scaled focal adhesion length (μm) ± SEM plotted as a function of substrate stiffness for scRNA (black), siPiezo1 (red), V− (continuous line) and V+ (dashed line) conditions.

forces exerted by scRNA and siPiezo1 MSCs on the different (soft, stiff, V−, V+) substrate types through traction force microscopy (TFM). Substrates were prepared using 200 nm fluorescent beads, the displacement of which was calculated before and after cell-applied deformations and then converted into forces. We note that using an elastic algorithm likely overestimates forces on viscoelastic V+ substrates, as it does not account for the dissipation of exerted tractions (Supplementary Data Fig. 4a–d). Nevertheless, standard TFM has proved to be a useful tool for investigating relative changes in cell-exerted forces on viscoelastic PAAm hydrogels[27]. Accordingly, we found that on soft substrates, scRNA MSCs exhibited increased traction force generation in conditions of increased substrate stress relaxation (V+), whereas this increased traction force generation was abrogated in siPiezo1 MSCs (Supplementary Data Fig. 4b). On stiff substrates, both siPiezo1 and scRNA MSCs generated significantly lower average traction forces as substrate stress relaxation increased (Supplementary Data Fig. 4c). This response was further reduced in siPiezo1 MSCs, highlighting the role of Piezo1 in traction force generation mechanisms within the cell, consistent with previous reports[16]. Perturbation studies also highlighted this relationship; where inhibiting cell contractility with blebbistatin induced similar cell-substrate interaction phenotypes as those produced by siPiezo1 MSCs (Supplementary Data Fig. 5a, b).

Overall, our results underscore the role of Piezo1 in mediating viscoelasticity-sensing in soft but not stiff ECMs, in which Piezo1 knock down lessened the cell response to increased substrate stress relaxation but did not fully inhibit it. Indeed, our experimental data agrees with the molecular clutch dynamics in response to matrix viscoelasticity that were first proposed by Chaudhuri and colleagues[9], in which clutch engagement is enhanced by substrate stress relaxation on soft environments but is inhibited above a stiffness threshold ($E$ - 1 kPa)[9,22]. Considering these results, we explored how Piezo1 mediates clutch dynamics in cells interacting with a viscoelastic substrate (Fig. 2e). We hypothesised that Piezo1-integrin concerted action is pivotal for clutch engagement in response to i) faster stress relaxation at low stiffness (soft V+) as well as ii) high stiffness conditions with slow stress relaxation (stiff V−). We consolidated this hypothesis by modifying the clutch model to introduce the effect of Piezo1 within the viscoelastic molecular clutch framework. By representing the coordinated action of Piezo1 and integrins via the modulation of integrin-ECM unbinding rates ($k_{off}$), we were able to reproduce experimental results. Model data thus supports our hypothesis that places Piezo1 as a key regulator of clutch engagement in response to viscoelastic cues, particularly in soft regimes.

### Matrix viscoelasticity and Piezo1 regulate downstream mechanotransduction and mitochondrial morphology

We next investigated whether the observed changes in clutch engagement as a function of Piezo1 expression and substrate viscoelasticity activated downstream mechanotransduction events. Increased cytoskeletal tension has been associated with nuclear compression and chromatin compaction[34,35]. Because of this, we assessed

and quantified nuclear projected spreading area (Fig. 4a, b, f). We observed that nuclear spreading was higher in scRNA MSCs cultured on soft, faster stress relaxation (soft V+) hydrogels compared to nuclear spreading in scRNA MSCs cultured on elastic (soft V−) hydrogels (Fig. 4a). Contrarily, nuclear spreading was reduced in scRNA MSCs cultured on stiff V+ hydrogels compared to stiff V− ones (Fig. 4b). Notably, in siPiezo1 MSCs, this increased nuclear spreading was abrogated on both soft V+ and stiff V− hydrogels (Fig. 4f), indicating that viscoelasticity and Piezo1-mediated cytoskeletal tension directly act on the nucleus.

It has recently been proposed[36,37] that the force applied to the nucleus could dictate the nuclear translocation of important transcription factors, such as Yes-associated protein (YAP), independently of other specific signalling pathways. Nuclear flattening (i.e. increased projected spreading area) increases nucleocytoplasmic transport, leading to the nuclear translocation of factors such as YAP through differential effects on active versus passive transport. Previous work has reported that Piezo1 activity is linked to enhanced YAP nuclear translocation[18] and on the importance of Piezo1 in sensitively sensing tensional changes to regulate nuclear size in response to exogenously applied shear stress[38]. Thus, we sought to understand whether the observed morphological changes in the nucleus reflected transcription factor translocation by assessing the localisation of YAP (Fig. 4c). In both siPiezo1 and scRNA MSCs, YAP translocated into the nucleus (YAPnuc/YAPcyto > 2) in response to increased substrate stiffness (Fig. 3d, e, g). However, on soft substrates, YAP was mostly cytoplasmic in siPiezo1 and scRNA MSCs, and did not translocate in response to increased stress relaxation (V+). YAP nuclear translocation in response to molecular clutch activation mechanisms has been shown to occur past an elasticity threshold of $E$ - 5 kPa[6]. Therefore, soft viscoelastic (soft V+) substrates did not promote sufficient adhesion maturity and reinforcement, or stress fibre formation to facilitate YAP nuclear localisation (nucYAP/cyto YAP < 2). These data support recent work that demonstrates that even if cells increase spreading area, adhesion reinforcement is needed to achieve YAP nuclear localisation[39]. When siPiezo1 MSCs were cultured on stiff substrates with faster substrate relaxation (stiff V+), both nuclear spreading area and YAP translocation were reduced. Similarly, on stiff V− substrates, siPiezo1 MSCs showed significantly lower levels of nuclear YAP (Fig. 4e). Likewise, when siPiezo1 MSCs were cultured on FN-coated glass, we observed smaller nuclei (Supplementary Data Fig. 6b) as well as slightly reduced YAP nuclear translocation (Supplementary Data Fig. 6a, c), when compared to scRNA MSCs. These results concur that regardless of the substrate employed, reducing Piezo1 expression generally decreases YAP nuclear translocation.

To further corroborate this relationship, plotting the mean nucYAP/cytoYAP against the mean nuclear spreading area across all experimental conditions revealed a strong positive correlation between the two variables (Fig. 4h, $R^2 = 0.9332$). These results highlight that YAP is sensitive to both the effects of viscoelasticity and Piezo1-

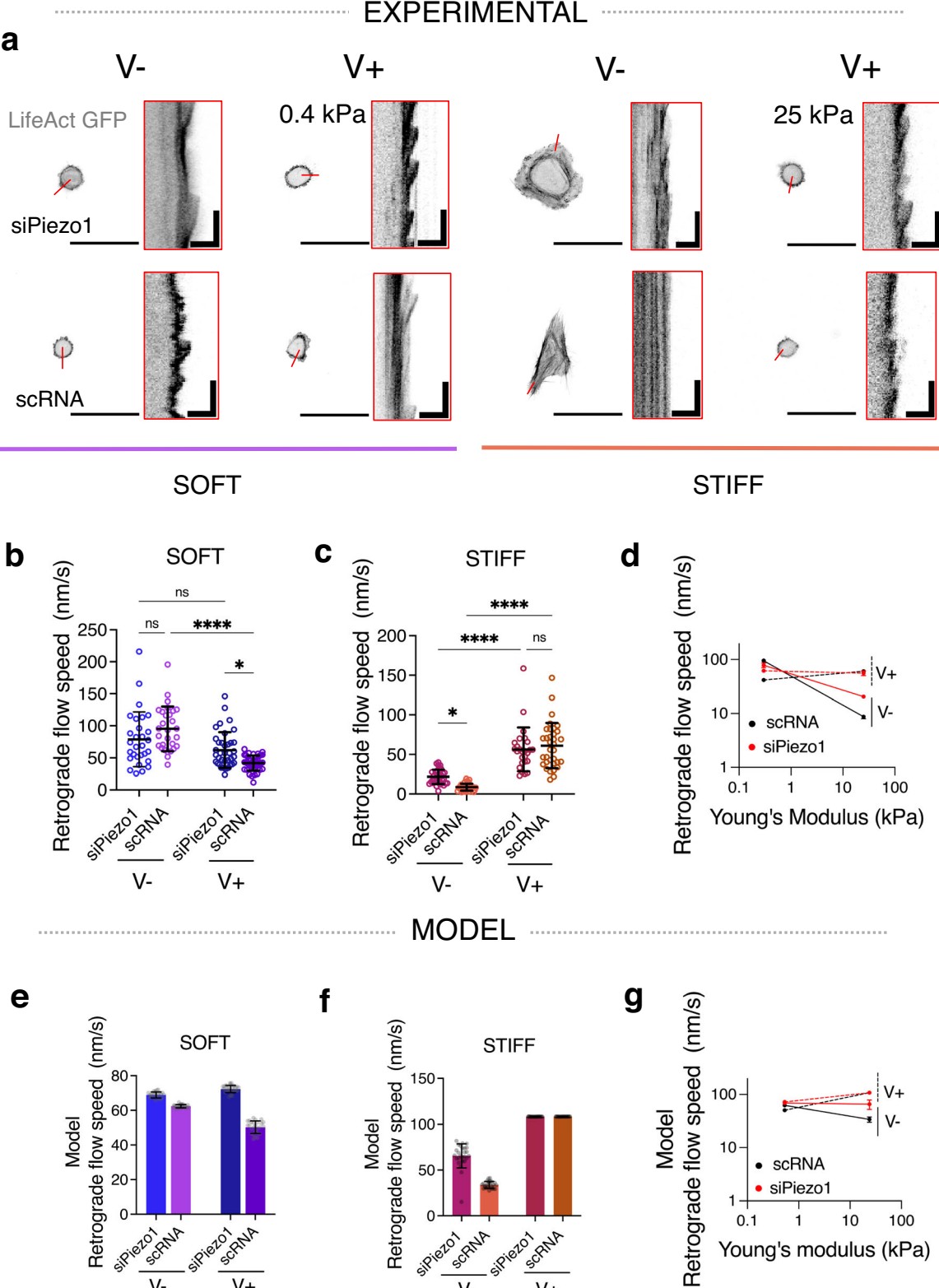

mediated cytoskeletal tension, which both act on the nucleus starting from the molecular clutch.

We then investigated how the cell-substrate interaction affected the cell's metabolic capacity, as recent evidence has linked stiffness-dependent integrin signalling to the modulation of mitochondrial activity[40]. We performed preliminarily oxygen consumption rate (OCR) experiments on stiff and soft Matrigel substrates (Supplementary Data Fig. 7a). We used Matrigel substrates in these experiments due to the

technical implications of using the Seahorse assay to measure OCR, which made implementing viscoelastic PAAm substrates complex owing to having to polymerise the different hydrogels on a coated 96 well plate format. By contrast, soft ($E \sim 0.2$ kPa) and stiff ($E \sim 1$ GPa) Matrigel systems have previously been applied into the seahorse assay experimental set-up[41]. OCR measurements indicated that Y201 MSC mitochondrial respiration rates were sensitive both to substrate stiffness and Piezo1 expression (Supplementary Data Fig. 7b–d). Conversely, non-

**Fig. 3 | Actin retrograde flow in response to matrix viscoelasticity is stiffness and Piezo1-dependent. a** Representative images of LifeAct GFP transfected siPiezo1 and scRNA MSCs cultured on the soft (left) and stiff (right) hydrogel groups. Scale bar = 50 μm. Red insets are kymographs showing the movement of actin features, scale bar = 5 μm (horizontal) and 60 s (vertical). **b, c** Quantified retrograde flow speed (nm/s) in siPiezo1 and scRNA MSCs cultured on the soft (left) and stiff (right) hydrogel groups. In (**b**) from left to right $n$ = 29, 29, 38 and 39 individual kymograph measurements from 13, 8, 16 and 9 cells; in (**c**) from left to right n = 35, 39, 26 and 34 individual kymograph measurements from 12, 16, 13 and 17 cells. Data from $N$ = 2 independent experiments. Data shown as individual values, mean ± SD. **d** Summary of mean experimental

retrograde actin flow speed (nm/s) ± SEM plotted as a function of stiffness for scRNA (black), siPiezo1 (red), V− (continuous line) and V+ (dashed line) conditions. **e, f** Model predictions of retrograde flow speed (nm/s) of cells on soft (left) and stiff (right) substrates. Data shown as individual values, mean ± SD ($N$ = 26 simulations). **g** Summary of model predictions of mean retrograde actin flow speed (nm/s) ± SEM plotted as a function of stiffness for scRNA (black), siPiezo1 (red), V− (continuous line) and V+ (dashed line) conditions. Statistical analyses were performed using a two-way ANOVA test. $P$ values indicating significance, ns > 0.05, *≤0.05, ***≤0.001, ****≤0.0001. Specific $p$ values and descriptive statistics are provided in the Source Data.

mitochondrial respiration was only significantly decreased when Piezo1 was knocked down on stiff Matrigel substrates (Supplementary Data Fig. 7e). We found that a soft matrix decreased cellular respiration capacity independently of Piezo1 expression (Supplementary Data Fig. 7a). However, on stiff substrates, mitochondrial-dependent respiration was increased, but this increase was abrogated if Piezo1 was knocked down (Supplementary Data Fig. 7e). Our data thus indicate that mitochondria are targeted by the Piezo1-modulated mechanosensing of matrix mechanics, regulating cellular respiration. Therefore, in order to assess mitochondrial respiration in response to varying viscoelasticity in our experimental set up, we labelled the outer mitochondrial membrane protein TOMM20 via immunostaining (Fig. 4i) and assessed mitochondrial elongation by quantifying the mean mitochondrial Form Factor in all experimental conditions (Fig. 4j–l). Mitochondrial elongation provides information on the fusion vs fission events in the organelle, alluding to the proliferative and respiratory state of the cell. In the past, mitochondrial fission and fusion has been shown to be significantly altered in response to changes in substrate stiffness[42]. Indeed, mitochondria appeared shorter on soft elastic (soft V-) compared to stiff elastic (stiff V-) substrates, as previously reported in ref. 40. Increased stress relaxation on soft hydrogels promoted mitochondrial elongation and abrogated matrix-induced mitochondrial fission (Fig. 4j). Whereas on stiff substrates (Fig. 4k), mitochondria were most elongated on scRNA cells cultured on slow stress relaxing substrates (stiff V−). In this case, both stress relaxation and Piezo1 knock down conditions decreased mitochondrial form factor. Besides mitochondrial elongation, we also quantified the total mitochondrial area in the cell, to estimate respiration rates[42]. Indeed, from total mitochondrial area data (Supplementary Data Fig. 8), it is possible to hypothesise that overall cell respiration is highly linked to mitochondrial mass inside the cell. This means that increased substrate relaxation (V+) enhances cellular respiration and metabolism in soft ECMs (Supplementary Data Fig. 8a), whereas on stiff matrices (Supplementary Data Fig. 8b), increased stress relaxation decreases overall cellular respiration and slows the cell's metabolic capabilities. Still, it does not significantly reduce mitochondrial form factor (Fig. 4k), suggesting that cells are not under oxidative stress. In terms of Piezo1's role in mitochondrial dynamics, it appears that its knock down generally promotes mitochondrial fission (Fig. 4l). This is most likely a consequence of the Piezo1's role in mediating the cell's interaction with its environment. Thus, mitochondrial fission induced by Piezo1 knock down is a read-out of the effects of Piezo1 on overall cell morphology.

## Matrix viscoelasticity and Piezo1's influence on transcriptomic phenotype

Finally, to monitor matrix viscoelasticity and Piezo1 activity-dependent transcriptional changes and to obtain reference transcriptomic phenotypes, we performed RNAseq on all experimental conditions at the 48 h timepoint of the previously shown mechanotransduction and metabolic data. This timepoint was chosen for two reasons: (i) to compare RNAseq data with previously shown data; and (ii) because in previous studies of MSC responses to varying stiffness, stress relaxation and ligand density in 3D matrices, RNAseq was performed at 40 h to ensure mature adhesions had formed with minimal cell proliferation[43,44]. Prior to assessing differential gene expression in

response to viscoelastic matrices, we ensured that Piezo1 knock down was efficient in cells used for sequencing, to control for experimental upscaling. For this, scRNA and siPiezo1 cells were cultured on FN coated glass and RNAseq was performed at the 48 h timepoint. Gene counts for Piezo1 were reduced in siPiezo1 cells, with a significant fold change after performing differential gene expression analysis (Supplementary Data Fig. 9a, b). Further, we saw a clear transcriptome change from scRNA to siPiezo1 cells, where 66 genes were differentially expressed ($p$ < 0.05) by the transient knock down of Piezo1 (Supplementary Data Fig. 9c). We also performed gene ontology (GO) enrichment analysis on the DE genes between de siPiezo1 and scRNA comparison. GO enrichment places top DE genes in functional modules of relevant sub-ontologies[45–47]. GO enrichment by over-representation analysis (ORA) highlighted cell adhesion as the top differentially expressed biological process (Supplementary Data Fig. 9d), supporting literature evidence of a Piezo1-integrin concerted action[14–17]. These findings also underpin the hypothesis of our work, which places Piezo1 as a key mediator of adhesion complexes, and thus of molecular clutch dynamics at the cell-ECM interface.

We then performed differential expression analysis on the resulting RNAseq data from both siPiezo1 and scRNA MSCs cultured on our hydrogel substrates and generated a curated heatmap with z-score up- and down-regulated genes plotted for each hydrogel stiffness range (Fig. 5a and Fig. 6a) as well as a general differential expression heatmap of genes with $p$ < 0.05. In the stiff hydrogel group, a total of 731 genes were differentially expressed across the V+ and V- conditions (Supplementary Data Fig. 10). Curated family heatmaps display z-score rlog-normalised expression for genes from five functionally defined relevant gene families: integrins, actin, YAP, mitochondria and contractility, which were used to create a visual representation of inner stiffness-group transcriptome variation. From these data it is possible to discern the strong interplay between Piezo1, integrins, actin cytoskeleton and myosin-based contractility mechanisms in the cell at this stiffness range (25 kPa); as Piezo1 knock down generally decreases gene expression within these functional families (Fig. 5a). Also, in accordance with our data, integrin, actin and contractility subfamilies are generally overexpressed in V− conditions, and levels decrease in response to enhanced energy dissipation (V+). Additionally, we assessed mitochondrial and YAP functional groups. The YAP subfamily is overexpressed in scRNA V−, with both Piezo1 knock down and energy dissipation having a downregulatory effect, as shown in Fig. 4e. Further, we observed that Piezo1 knock down seems to produce a decrease in mitochondrial genes, as demonstrated experimentally in Fig. 4k. Finally, a general downregulation of functional gene families is observed in response to enhanced energy dissipation at this stiffness.

We expanded our analysis to perform GO enrichment between experimental conditions. First, we assessed the enrichment of the DE genes from scRNA cells seeded on stiff elastic (V−) versus viscoelastic (V+) matrices to assess the processes that drive Y201 MSCs' response to viscoelasticity at this stiffness (Fig. 5b). The top sub ontologies to emerge from this GO analysis were cell motility, bone remodelling and G-protein-coupled receptor (GPCR) signalling. These are processes that are associated in the literature with cell interactions in stiff 2D viscoelastic substrates. For example, our group has demonstrated that

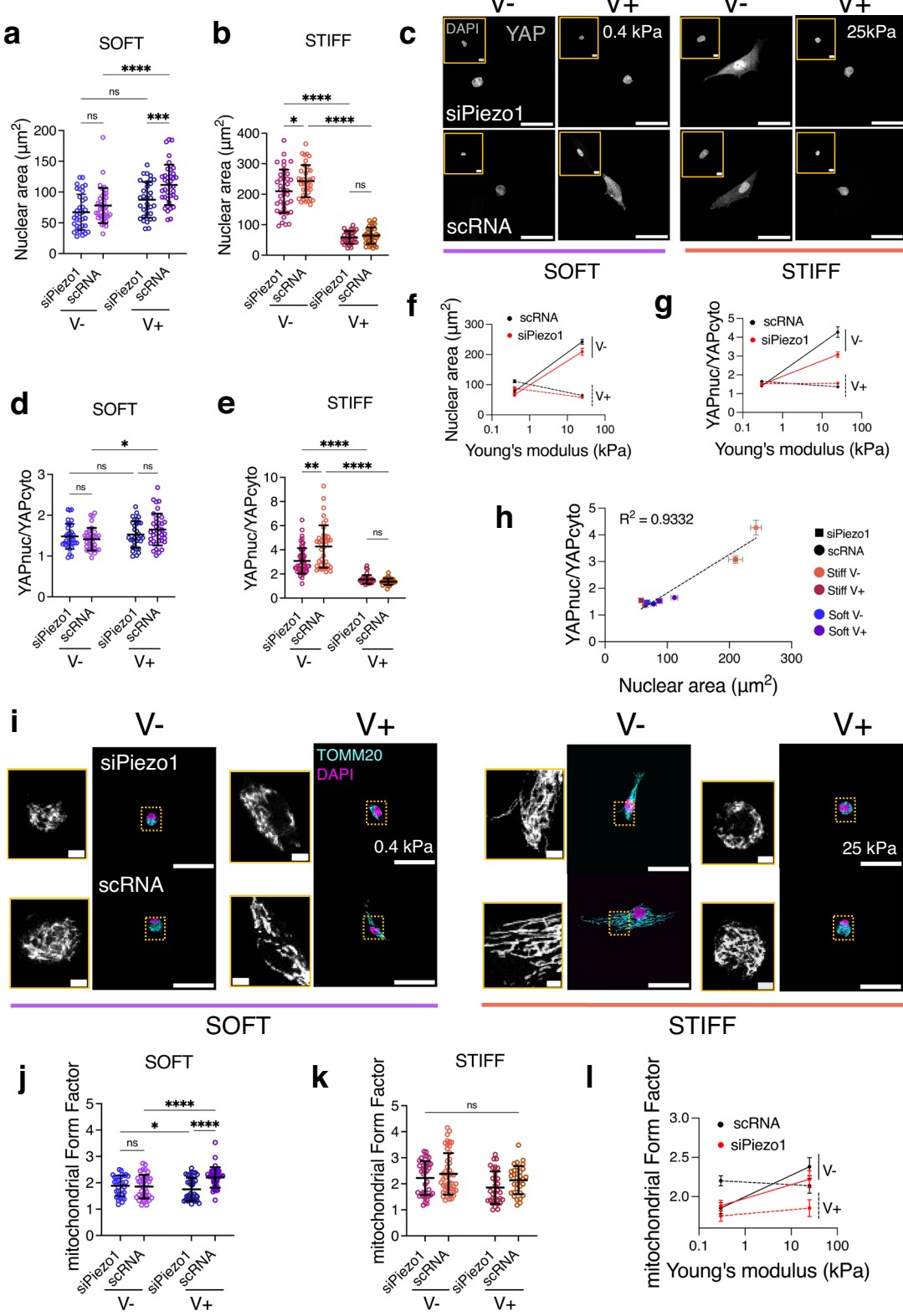

cell motility is reduced in breast epithelial cells in response to enhanced substrate stress relaxation at $E > 1$ kPa[48]. Similarly, at stiffness $E > 1$ kPa, we have shown that viscoelasticity reduces cell spreading area, molecular clutch engagement and YAP nuclear translocation, which are all processes involved in bone remodelling. In previous work by our group, we have demonstrated that substrate stress relaxation in hydrogels of similar stiffness ($E \sim 13$ kPa) promoted chondrogenesis in hMSCs[25]. Finally, the GPCR pathways ontology includes genes that

belong to the Ras-subfamily of GTPases (*RASD1*) or Rho signalling (*GNAT1*), which mediate processes that have been previously highlighted in studies describing cell response to 2D viscoelastic substrates[24].

We performed the same comparison on siPiezo1 MSCs to investigate Piezo1-independent transcriptional mechanisms of matrix viscoelasticity for the stiff substrate. In this GO analysis, processes such as hormone secretion and regulation, muscle tissue development and

**Fig. 4 | Matrix viscoelasticity and Piezo1 expression regulate downstream mechanotransduction and mitochondrial morphology. a, b** Quantified nuclear spreading area on the soft (left) and stiff (right) hydrogel groups. In (**a**) from left to right, $n = 36, 37, 32$ and $38$ cells; in (**b**) from left to right, $n = 39, 38, 31$ and $35$ cells. Data from $N = 3$ independent experiments. Data shown as individual values, mean ± SD. **c** Representative images of nuclei (insets) and YAP in siPiezo1 (top) and scRNA Y201 MSCs (bottom) cells on soft (0.4 kPa, left) and stiff (25 kPa, right) hydrogel groups of varying stress relaxation. Scale bar = 50 μm, DAPI inset scale bar = 5 μm. **d, e** Quantified nuclear over cytoplasmic YAP (nucYAP/cytoYAP) ratio on the soft (left) and stiff (right) hydrogel groups. In (**d**) from left to right, $n = 34, 34, 32$ and $38$ cells; in (**e**) from left to right $n = 55, 40, 34$ and $31$ cells from $N = 3$ independent experiments. Data shown as individual values, mean ± SD. **f** Summary of mean nuclear spreading area ± SEM plotted as a function of stiffness for all conditions. **g** Summary of mean nucYAP/cytoYAP ratio ±SEM plotted as a function of stiffness for all conditions. In (**f, g**) conditions are as follow: scRNA (black), siPiezo1 (red), V− (continuous line) and V+

(dashed line). **h** Correlation between nucYAP/cytoYAP and nuclear spreading area for all conditions. scRNA conditions have rounded symbols whilst siPiezo1 are squared. Symbol colour indicates stiffness and energy dissipation conditions. Data shown as mean ± SEM. $R^2$ obtained from simple linear regression analysis fitting mean values **i** Representative images of siPiezo1 (top) and scRNA (bottom) MSCs immunostained for TOMM20 (cyan) and DAPI (magenta) cultured on soft (0.4 kPa, left) and stiff (25 kPa, right) hydrogels for 48 h. Scale bar 50 μm; TOMM20 inset scale bar 2 μm **j, k** Quantified mean mitochondrial form factor of MSCs cultured on the soft (left) and stiff (right) hydrogel groups. In (**j**) from left to right, $n = 32, 40, 40$ and $36$ cells; in (**k**) from left to right, $n = 37, 48, 33$ and $32$ cells. Data from $N = 3$ independent experiments. **l** Summary of mean mitochondrial form factor ± SEM plotted as a function of stiffness for all conditions. **a, b, d, e, j, k** Statistical analyses were performed using a two-way ANOVA test. $P$ values indicating significance, ns > 0.05, *≤0.05, **≤0.01, ***≤0.001, ****≤0.0001. Specific $p$ values and descriptive statistics are provided in the Source Data.

---

differentiation and smooth muscle cell migration were highlighted (Fig. 5c). From these results, we hypothesise that Piezo1 expression is crucial for mediating viscoelasticity-induced changes in bone remodelling and locomotion processes. However, when Piezo1 is silenced in Y201 MSCs, enhanced substrate stress relaxation affects myosin-related ontologies (i.e. muscle tissue development and differentiation). These data are supported by our perturbation studies (Supplementary Data Fig. 5), where inhibiting Myosin-II mediated cell contractility with blebbistatin inhibited cell mechanoactivation to similar levels to those of Piezo1 knock down. Results thus suggest that myosin signalling mechanisms still respond to enhanced viscoelasticity, but that Piezo1 is required to further these adaptions into canonical mechanotransduction pathways, as highlighted in Fig. 5b. This concomitant action between myosin signalling and Piezo1 has also been highlighted previously[16], and the interplay between Piezo1 and cell-generated forces is key to allow for the spatial segregation of mechanotransduction events such as locomotion, bone remodelling and GPCR signalling.

Finally, we compared transcriptomic data from scRNA vs siPiezo1 MSCs cultured on stiff elastic (25 kPa, V−) matrices (Fig. 5d). The top sub ontologies to emerge from this GO analysis of the top DE genes included cardiovascular system remodelling, bone remodelling and morphogenesis. Piezo1 knockdown downregulated genes such as *BMP2*, *VEGFA*, *ITGB3*, which are involved in cell proliferation, differentiation and integrin-related signalling. In fact, these processes have been directly associated with Piezo1's activity and expression in MSCs[49], and support our hypothesis that Piezo1 knock down reduces overall molecular clutch engagement (Fig. 2c and Fig. 3d) on stiff elastic matrices and glass substrates (Supplementary Data Fig. 2c, d and Supplementary Data Fig. 9d), as well as reducing YAP nuclear translocation (Fig. 4g and Supplementary Data Fig. 6c), which has been shown to regulate bone differentiation in MSCs[50].

We conducted the same analysis for scRNA vs siPiezo1 MSCs cultured on soft substrates, and identified 177 DE genes (Supplementary Data Fig. 11), which suggests that when MSCs are cultured on soft substrates both Piezo1 expression and substrate energy dissipation produce a less pronounced effect on the transcriptome than when MSCs are cultured on stiff substrates. A curated heatmap of rlog normalised gene expression across relevant gene families was plotted (Fig. 6a). Here, in the scRNA V+ conditions, genes such as vinculin (*VCL*), integrin β1 and β3 (*ITGB1* and *ITGB3*), Myosin11 (*MYH11*) and *ROCK2* show overexpression; highlighting their role in mediating the mechanoactivation we observed experimentally in response to enhanced energy dissipation at this stiffness. Accordingly, these all became downregulated in siPiezo1 cells. Additionally, increased energy dissipation on scRNA MSCs upregulated *YAP1* and *TEAD* associated genes, commensurate with data shown in Fig. 4, d, where we show that although YAP does not become nuclear at this stiffness in response to increased energy dissipation, there is an increase in

mechanotransductive mechanisms. Finally, in our mitochondrial gene family, two mitochondrial biogenesis genes (*TFAM* and *NDUFS1*) were overexpressed in scRNA V+ conditions, indicating that enhanced energy dissipation at this stiffness potentially increases the metabolic capacity of cells without inducing mitochondrial membrane damage; this is in line with data we showed in Fig. 4j. and Supplementary Fig. 8.

We then assessed and compared enriched DE genes in scRNA MSCs cultured on soft elastic (soft V−) versus on viscoelastic (soft V+) matrices (Fig. 6b). The top sub ontologies to emerge from this comparative GO analysis were the GPCR pathway, synaptic signalling and central nervous system development. Interestingly, in the central nervous system development subontology, we found the Wnt family genes, *WNT7A* and *WNT3*, which are important regulators of YAP and respond to matrix stiffness[51]. Although at this stiffness range (−0.4 kPa), viscoelasticity did not promote YAP nuclear translocation in cells (Fig. 4f), it is possible that components of the non-canonical Wnt signalling pathway[52], such as *WNT3* and GPCRs (Gα subunits), still become activated to support cell proliferation in response to enhanced substrate stress relaxation. These sub-ontologies have been highlighted in previous studies that investigated transcriptional changes in MSCs encapsulated in dynamic viscoplastic matrices compared to fully crosslinked elastic matrices[53]. Additionally, we found that the synaptic signalling sub-ontology was underscored in the enrichment analysis. Indeed, genes in this subontology include phospholipase C beta 1 (*PLCB1*), which is associated with cytoskeletal rearrangement processes in gastric tumour tissue samples[54] and with other genes that encode $Ca^{2+}$ responsive channels (such as *CACNA1E*, *CACNA1B* and *KCNMB1*). These results could explain our data on scRNA MSC enhanced cell spreading area (cytoskeletal rearrangement) and clutch engagement (Fig. 1g; Fig. 2b and Fig. 3b) in response to increased viscoelasticity on soft matrices.

As before, we compared transcriptomic data from siPiezo1 MSCs cultured on soft elastic (soft V−) vs on soft viscoelastic (soft V+) matrices to highlight the Piezo1-dependent cell response to viscoelasticity at this stiffness. Notably, our DE gene enrichment analysis showed that most of the previously identified GO sub-ontologies were replaced by cell division processes, which were mostly up regulated in viscoelastic hydrogels in a Piezo1-independent manner. In our data, we do not see any phenotypic difference between siPiezo1 MSCs cultured on soft elastic (V−) vs viscoelastic (V+) hydrogels. Nonetheless, it might be that viscoelasticity in soft matrices promotes cell proliferation, as previously reported in a cancer cell line[9] and this may happen independently of Piezo1.

We compared transcriptomic data from scRNA and siPiezo1 MSCs cultured on soft viscoelastic (V+) hydrogels. Particularly, Piezo1 knock down promoted the down regulation of several ontologies relating to immune response regulation and cell-cell adhesion. MSCs have important immunomodulatory properties, which are responsive to

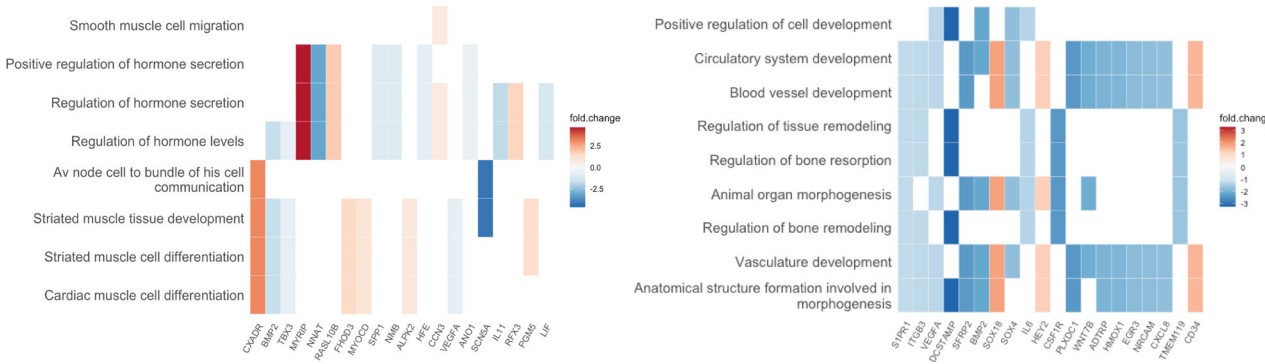

**Fig. 5 | RNA-seq analysis of siPiezo1 and scRNA Y201 MSCs cultured on stiff elastic (V−) and viscoelastic (V+) matrices. a** Curated z-score (rlog-normalised expression) heatmap of stiff group genes across experimental conditions per defined gene families (Integrins, Actin cytoskeleton, YAP signalling, Mitochondria and Contractility). **b** (Top) Heatmap of enriched results from Over Representation Analysis (ORA), which overlap with the most DE genes in the scRNA MSC V− vs scRNA MSC V+ comparison. (Bottom) A network of the three top enriched DE gene groups connected by their overlapping genes. **c** Heatmap of enriched results from ORA that overlap with the most DE genes in the siPiezo1 V− vs siPiezo1 V+ comparison. **d** Heatmap of enriched results from ORA that overlap with the most DE genes in the scRNA V− vs siPiezo1 V− comparison.

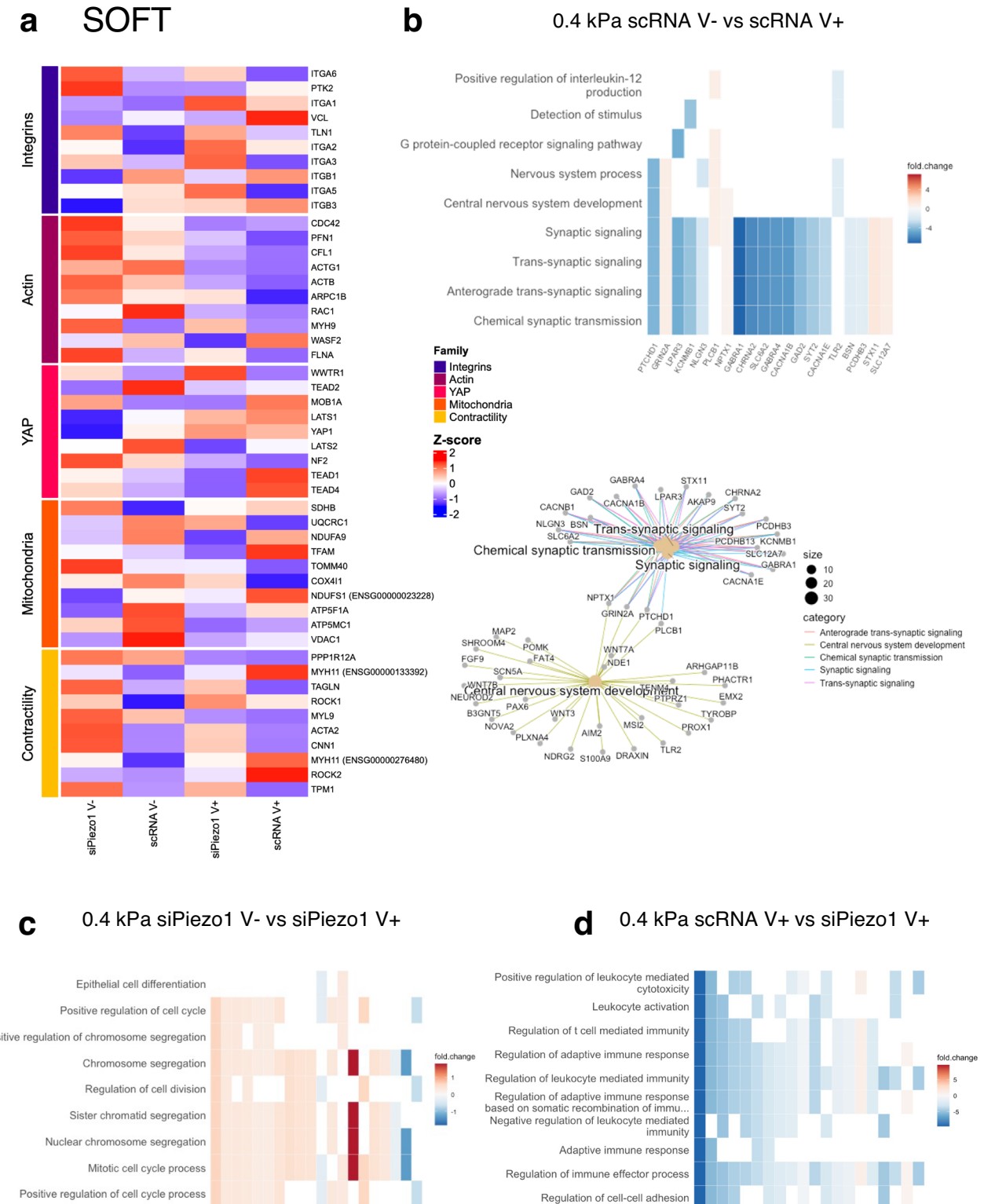

**Fig. 6 | RNA-seq analysis of siPiezo1 and scRNA Y201 MSCs cultured soft elastic (V−) and viscoelastic (V+) matrices. a** Curated z-score (rlog-normalised expression) heatmap of soft group genes across experimental conditions per defined gene families (Integrins, Actin cytoskeleton, YAP signalling, Mitochondria and Contractility). **b** (Top) Heatmap of enriched results from Over Representation Analysis (ORA), which overlap with the most DE genes in the scRNA MSC V− vs scRNA MSC V+ comparison. (Bottom) A network of the top enriched DE gene groups connected by their overlapping genes. **c** Heatmap of enriched results from Gene Set Enrichment Analysis (GSEA) that overlap with the most DE genes in the siPiezo1 V− vs siPiezo1 V+ comparison. **d** Heatmap of enriched results from ORA that overlap with the most DE genes in the scRNA V− vs siPiezo1 V− comparison.

matrix mechanics[55,56]. Similarly, Piezo1 mechanosensing has recently been linked to adherens cell-cell junction formation in endothelial cells[57,58], as well as to immune cell migration[59]. However, the link between Piezo1 knock down and MSC downregulation of immuno-modulatory genes remains unclear, and something that should be addressed in future work. Interestingly, in previous analyses of transcriptomic changes in MSCs encapsulated in 3D hydrogels of increasing stiffness (in which $E$ increased from 3 kPa to 18 kPa), several immunomodulatory markers were differentially expressed[43]. This was found to be linked to stiffness-induced activation of the immune and inflammatory transcription factor NFkB-p65. Previous work from our group[55,60] has addressed that MSCs reach an immunomodulatory phenotype in intermediate adhesion states, where adhesion – and thus- molecular clutch mechanisms are active but not so overly engaged to promote differentiation; from these results we propose that Piezo1 is a mediator of stiffness-induced immunomodulation in MSCs on softer substrates.

Overall, our transcriptomic analyses indicate that both viscoelasticity and Piezo1 activity regulate a diverse range of processes in MSCs, and that depending on substrate stiffness, different gene groups respond to changes in matrix viscoelasticity as well as to Piezo1 knock down.

## Discussion

Previous studies have shown that cells respond to matrix viscoelasticity via dynamic molecular clutch mechanisms that sense and engage with the cell's mechanical microenvironment in a stiffness-dependent way[9,22]. Here, we demonstrate that Piezo1 is an important sensor of soft matrix viscoelasticity. Piezo1 knock down in the MSC cell line Y201 results in a mechanobiologically impaired phenotype that is characterised by reduced cell spreading behaviour, as wells as reduced molecular clutch engagement, mechanotransduction and mitochondrial metabolic activity. In addition, we characterise how cells respond to viscoelasticity in a stiffness-dependent manner, where on soft ($E \sim 0.4$ kPa) substrates, faster stress relaxation promotes cell mechanoactivation while on stiff ($E \sim 25$ kPa) substrates, the opposite is true[9]. From our results, we propose that cell mechanoactivation in response to faster stress relaxation in soft matrices is abrogated when Piezo1 is knocked down, highlighting the role of this channel in relaying viscoelasticity at this stiffness. Increased stress relaxation on stiff substrates prevented mechanoactivation—reported also recently by other groups[61]. We therefore suggest that Piezo1 is not able to influence cell response in a stiff viscoelastic regime, as it cannot potentiate adhesion complexes if integrins are not first recruited. To confirm our experimental observations, a modified molecular clutch model was here devised, which accounted for the interaction of the cell with a viscoelastic substrate as well as for the concerted action of Piezo1 and integrins in clutch engagement. The modified model recapitulated previously observed stiffness-dependent clutch engagement in a viscoelastic (dissipative) context, where low stiffness ($E \sim 1$ kPa) promoted clutch engagement[9,22] and high stiffness blunted mechanosensing[61]. Most importantly, simply by considering an effect in integrin activation, simulations performed with the modified model captured how Piezo1 channel knockdown generally decreased clutch engagement and abrogated clutch potentiation in response to enhanced stress relaxation in soft matrices. Finally, by performing RNAseq, we obtained transcriptomic phenotypes of cell-substrate interaction, revealing gene signatures that highlight how cells respond to substrate viscoelasticity at two different stiffnesses and in terms of Piezo1 expression.

Our findings extend our current understanding of cell responses to substrate viscoelasticity and place the mechanosensor Piezo1 as a key mediator of these cues in soft substrates. Soft tissues in the body, such as brain, lung tissue or bone marrow, exhibit the highest degree of viscoelasticity[8]. Coincidentally, Piezo1 has been implicated in the physiological regulation and pathophysiology of the central nervous system, being originally identified in a mouse neuroblastoma cell line[12] and has been shown to regulate the differentiation of neural pluripotent stem cells[18]. Indeed, many of the differentially expressed genes in the soft group comparison (with the soft substrate used in this study emulating the stiffness of the human brain[62]), included genes linked to the central nervous system and synaptic transmission processes (Fig. 6b). While our study was conducted in an MSC line, it opens the door to future studies of how viscoelasticity and Piezo1 mediate cell behaviour in a more specific physiological context, such as neural tissue.

The soft viscoelastic matrices investigated here provide a suitable platform with which to further our understanding of tissue physiology, where the dissipative properties of native ECMs can be recapitulated in vitro, independently of stiffness. These findings also point towards the development of different tools for molecular cell-ECM interaction studies, mechanotransduction and transcriptomics to address tissue-specific questions.

## Methods

### PAAm hydrogel synthesis

Hydrogels were polymerised on clean borosilicate 12 mm diameter glass coverslips (VWR). Coverslip surfaces were functionalised with 3-(Acryloyloxy)propyltrimethoxysilane (Alfa Aesar). Hydrogel solutions prepared using stock solutions of 40% Aam (sigma) and 2% bisAam (sigma) mixed in different ratios for each hydrogel composition (Supplementary Data Table S2). For hydrogels requiring the addition of linear acrylamide, this was prepared beforehand by mixing 50ul of 40% acrylamide in 317 μl dH2O and polymerising it with 25 μl 1.5% TEMED and 8 μl 5% APS for 2 h at 37°, making a final stock concentration of 5% linear acrylamide, which was then added to the soft V+ hydrogel solution, as described in Supplementary Data Table S2. Once mixed, all hydrogel solutions were thoroughly mixed and vortexed prior to use. A hydrogel solution drop of 12 μl was placed on top of a hydrophobic (RainX treated) coverslip and an (acryloyloxy)propyl-trimethoxysilane treated coverslip was placed on top of the drop to synthesise flat hydrogels. Gelation was allowed to occur at RT for 30 min before detaching and swelling in PBS overnight at 4 °C.

To promote cell adhesion, substrates were functionalised with full length FN. This was done by placing 0.2 mg/ml sulfo-succinimidyl-6-(4-azido-2-nitrophenyl-amino) hexanoate (sulfo-SANPAH) (Thermo Fisher) in 0.5 mM pH 8.5 HEPES buffer onto the hydrogel surface and irradiating with ultraviolet (UV) light (365 nm) at a distance of 3 inches for 20 min. The darkened sulfo-SANPAH solution was removed, and substrates were rinsed twice with HEPES buffer and incubated with 10 μg/ml of FN in HEPES at 37 °C, overnight. All substrates were exposed to UV light in a sterile culture hood for 30 min. prior to use. Before plating cells, hydrogels were equilibrated in cell culture medium for 30 min. at 37 °C.

### Mechanical characterisation of hydrogels

Nanoindentation measurements were performed using a nanoindentation device (Chiaro, Optics11 Life) adapting a previously reported approach[63]. Measurements were performed at RT in PBS unless stated otherwise. To obtain the Young's modulus of the substrates, single indentation curves ($n > 75$) were acquired at a speed of 2 μm/s over a vertical range of 10 μm, changing the (x, y) point at every indentation. The selected cantilever had a stiffness of 0.52 N/m and held a spherical tip of 27.5 μm radius. A minimum of three indentation maps of 25 curves per replicate were measured. All collected curves were pre-processed and analysed with a custom-made graphical user interface, the analysis and all software used are described in detail in ref. 63 and available in ref. 64.

To assess substrate topography, nanoindentation load-displacement curves were analysed using the Prova software

(Optics11). Topographical measurements were derived from the z-axis adjustments made by the software's contact point detection algorithm between sequential indentations.

To perform stress relaxation measurements, at least 95 indentations were performed, each spaced at least 50 μm from the previous. The selected cantilever had a stiffness of 0.52 N/m and held a spherical tip of 27.5 μm radius. For each indentation, the probe moved at a strain rate of 5 μms$^{-1}$ until it reached an indentation depth (h) of 3 μm, which was maintained for 60 s using the instrument's closed feedback *Indentation control mode*. The applied strain ($\varepsilon_0$) for the stress relaxation measurements was calculated as: $\varepsilon = 0.2 \cdot a/R$, where $a = \sqrt{h \cdot R}$, where h is the indentation depth and R the probe radius[65]. Therefore, with an indentation depth of 3 μm and a probe radius of 27.5 μm, the applied strain was approximately 7%.

Acquired data was pre-processed using a previously published open-source software (time branch of the project)[64]. To analyse the stress relaxation behaviour of the material, an analysis script in the form of a jupyter notebook was developed[66]. Briefly, force-time F(t) curves were first aligned to zero force if their baseline was negative. Then, the maximum of F(t) and its corresponding time was found, yielding the point (t0, F0). Curves were therefore aligned to 0 time by a horizontal shift equal to t0. Following this, the signal was cropped between t0 and the maximum time before retraction, i.e. only the part of the signal where the indentation was kept constant was retained. Following this, F(t) was normalised by dividing the whole signal by F0. Because individual curves were too noisy to be analysed, an average curve was found and used for quantification of the time for the stress to decrease to an 80% of the original value as well as the energy dissipation of the materials. This was done by extracting the time at which force reaches 80% of original value.

To perform rheology measurements, hydrogels were prepared in 15 mm diameter PDMS moulds using 250 μl volumes. Samples were left to swell overnight and subsequently measured with a Physica MCR 301 rheometer (Anton Parr). The linear viscoelastic region was determined by performing amplitude sweeps from 0.01 to 10 % strain and then a strain of 0.1% was used to obtain frequency sweeps from 100 to 1 rad/s.

## Y201 hMSCs culture and transfection

Y201 hMSCs were grown in as adherent cultures in high glucose DMEM (Gibco), 10% FBS and 1% P/S. Cells were passaged every 3 days and used between passages 70-90.

The Piezo1 channel was routinely knocked down prior to experiments using siRNA. This was done with the pre-validated siRNAs library from Thermo Fisher. Specifically, siRNA1 (ID 138387), siRNA 2 (ID 138388) and siRNA 3 (ID 138389) were first screened and siRNA1, then referred to as siPiezo1, was thereafter routinely applied. A validated control siRNA Ambion™ Silencer™ Negative Control #1 was also used to ensure transfection efficiency and validate the channel knock down (scRNA). All siRNA was resuspended in Nuclease Free water (provided with kit) at 100 μM and stored at −80 °C. Prior to transfection, cells were plated on cell culture treated six well plates. Cells were left to adhere in antibiotic free supplemented media. After adhering for 24 h, media was changed to OptiMEM (Gibco) reduced serum medium and siRNA was introduced with Lipofectamine RNAiMax transfection reagent (Thermo Fisher), as per manufacturer's instructions. Briefly, 25pMol of siRNA/scRNA and 7.5 μl of RNAiMax lipofectamine were used per well and incubated for 72 h. After incubation, media was changed to supplemented high glucose DMEM (Gibco) and cells were left to recover overnight prior to use for experiments. For pharmacological inhibition of mechanosensitive ion channels, GsMTx4 (Cambridge Bioscience) was added to the cells for 24 h in 1% serum media at a concentration of 10 μM. For performing actin flow experiments, cells were transfected with a LifeAct-GFP plasmid (Ibidi) using a neon electroporator system (Invitrogen). 5 μg plasmid were used per 100 μl electroporator tip, as per manufacturer's instruction.

(−)-Blebbistatin (Sigma) was used at a concentration of 50 μM for myosin-II inhibition experiments. Cells were treated for 1 h prior to fixing.

## RNA extraction

RNA extraction was performed using a commercially available kit (RNeasy Micro kit, Qiagen). RNA concentration and quality was monitored by spectrophotometry using the Nanodrop 2000 (Thermo Scientific).

## RT-qPCR

For quantifying gene expression, real time quantitative Polymerase Chain Reaction was performed (RT-qPCR). 300–500 ng of RNA was used to generate complementary DNA (cDNA) using QuantiTect Reverse Transcription Kit Reagents (Qiagen). cDNA was amplified using Quantifast SYBR green qPCR kit (Qiagen) with specific primers for Piezo1 and ribosomal protein L3 (*RPL3*), which was used as a genetic internal control. Expression was quantified using the $2^{-\Delta\Delta Ct}$ method and amplification was carried out using an Applied Biosystems 7500 Real Time PCR system (Thermo Fisher).

## In cell western

For In-cell western Piezo1 protein quantification cells were seeded on a 48 well plate and cultured for 24 h post siRNA transfection. Three cell conditions were assessed: scRNA control, siPiezo1 and untransfected Y201 MSCs. After 24 h, cells were fixed for 15 min with 4% paraformaldehyde with 1% sucrose and subsequently permeabilised with permeabilising buffer (10.3 g Sucrose, 0.292 g NaCl, 0.06 g MgCl2 (hexahydrate), 0.476 g HEPES and 0.5 mL Triton-X100, in 100 mL, pH 7.2). Samples were washed with 3x with PBS- and blocked with a 1% milk protein PBS solution for 1.5 h on an orbital shaker at room temperature. The primary antibody was incubated in blocking solution overnight at 4 °C. The next day, samples were washed 3x with PBS and the secondary antibody was added at a dilution of 1:800 in blocking buffer (1:500 in the case of the CellTag for normalising the protein signal). Samples were then washed 5x with PBS and dried overnight in a chemical flow hood. Protein signal was measured with an Odissey Scanner at 700 and 800 nm once samples were dried. The intensity of the wells was then normalised to the cell-tag intensity and then normalised to the intensity of the samples which were only stained with the secondary antibody, to correct for background signal.

## Immunofluorescence staining

Samples were fixed with 4% formaldehyde in PBS for 15 min at room temperature (RT). 0.1% Triton X-100 in PBS was used to permeabilise cells for 10 min at RT. Samples were blocked in 1% BSA in PBS and incubated for 1 h and then the primary antibody was either incubated for 1 h at RT or overnight at 4 °C in an incubation chamber. Samples were washed with 0.1% Triton X-100 in PBS and the secondary antibody was incubated in 1% BSA in PBS for 1 h at RT. Finally, samples were washed thrice with 0.1% Triton X-100 in PBS and once in PBS. For image acquisition, samples were mounted on glass slides with Vectashield Hardset Antifade mounting medium with dapi.

See Supplementary Data Table S3 for antibody dilution and manufacturer.

## Actin flow measurements

On the day of imaging, LifeAct-GFP transfected cells were cultured on of the desired surfaces and left to adhere for a minimum of 2 h. Cells were then placed on the heated stage of the LSM980 Zeiss confocal fluorescent system. Imaging was performed at 37 °C in CO2 independent media (Gibco) with a 40x oil immersion objective with a numerical aperture of 1.3. Cells were illuminated with a 488 laser and images were acquired at a frame rate of 1 image every 2 s for a total of 4 min.

## Traction force microscopy

Cells traction forces were measured using the EVOS M700 (Thermo Fisher) imaging system at 20X magnification. Cells were seeded on glass petri-bound hydrogels prepared with 1 μl/ml of 0.1% FluoSpheres Carboxylate-modified microspheres (0.2 μm, 580/605, 2%) (Thermo Fisher) aqueous solution. In each sample, a total of 6 positions were tracked and z-stacks were taken (an image was taken every 5 μm) of both the cells and beads channels. Then, cells were detached with Trypsin for 10 min. and the same positions were imaged without cells to obtain the reference image.

## Oxygen consumption rate (OCR) measurements

OCR measurements were performed with the Agilent Seahorse XF Cell Mito Stress Test kit with an XF24 extracellular flux analyser (Seahorse Bioscience). XF24 plates were coated with either 23 μl of 100% Matrigel (soft) or 100 μl of 2% Matrigel (stiff). The solutions were evenly distributed with a flat pipette tip and the Matrigel was left to gel for 10 min. at 37 °C. 30,000 cells were seeded per well 24 h before performing measurements. The experiment was conducted according to the manufacturer's instructions with at least 4 technical replicates per experiment.

## RNA sequencing

RNA-Seq was performed by Glasgow Polyomics at the University of Glasgow. Briefly, strand-specific RNA-Seq libraries were prepared using the Illumina Stranded mRNA (poly A selected) library preparation kit with a NextSeq2000 system. In total, 100 × 100 bp paired end reads and an average of 30 M total reads were generated for each sample. All libraries were aligned to the Homo-sapiens.GrCh.cdna genome using Kallisto 0.46.1[67]. To perform differential gene expression, aligned reads wereyes imported to the DESeq[68] bioconductor package for R studio with the Txtimport function. Then, differential expression analysis was performed, and the counts matrix was obtained. To define relevant gene families, we curated a set of representative genes for each of five functional groups: Integrins, Actin cytoskeleton, YAP/Hippo, Mitochondria and Contractility, based on Reactome and KEGG annotations (see Supplementary Data Table S4). Genes were mapped to Ensembl Ids using biomaRt and expression values were averaged across experimental conditions. Z-score normalisation was applied across conditions. The resulting matrix was plotted using the ComplexHeatmap R package. Additionally, heatmaps of differential gene expression were obtained with the pheatmap function. GO terms analysis and graphs were obtained with the Interactive Enrichment Analysis tools implemented by the Gladstone Institutes Bioinformatics Core[47]. Enrichment analysis and resulting maps were created with go_hs_20241115 database. Raw RNA sequencing data and processed count matrices have been deposited in the Gene Expression Omnibus with the accession codes GSE288423, GSE288464 and GSE288465 for stiff, soft and glass experiments, respectively.

## Image analysis

**FN functionalisation analysis.** Immunofluorescent FN hydrogel images were opened using Image J 2.14.0 v (National Institute of Health, US). A square region of interest (ROI) was measured on three parts of the image, for all hydrogel conditions. A negative control (immunostained hydrogel without any FN functionalisation) was used to normalise background signal. The reported FN intensity values were therefore obtained by measuring the average integrated density per sample minus the negative control integrated density.

**Cell/nuclear morphology analysis.** Actin cytoskeleton images were converted to 8-bit, background was subtracted (rolling radius = 300) and a Gaussian blur of sigma 1 was applied. After this pre-processing, images were thresholded using Otsu's method. Thresholded features were selected with the Wand function and the resulting ROIs were measured to quantify parameters such as cell and nuclear area and circularity.

**Focal adhesion analysis.** FA quantification was performed with an adapted version of the Horzum protocol, previously described in literature[69]. Briefly, the vinculin channel images were cropped so that individual cells were in each image to analyse. Images were converted to 8-bit and background was subtracted (rolling radius = 50). Then, a Gaussian Blur (sigma = 1) was applied, and the contrast was enhanced with the CLAHE plugin (blocksize = 19, histogram = 256, maximum = 3, mask = None), the mathematical exponential was applied to further minimise background and the brightness and contrast was adjusted automatically (saturated = 0.35). The log 3D filter was applied (sigmax = sigmay = 3), the LUT was inverted and the image was thresholded using the Triangle method. The scale was adjusted according to the image pixel size calibration, adhesions were split with a watershed algorithm and finally the particles were analysed. Only particles over 0.75 μm² were measured. The results were saved for individual adhesions as well as a summary of all particles in the cell.

**Actin flow analysis.** Actin retrograde flow speed was calculated by kymograph analysis. In summary, timelapses were loaded onto image J and the area at the cell edge was sliced, producing a kymograph of line width 1 which plotted displacement over time using the Multi kymograph function. Flow speed was calculated by measuring and diving the bounding rectangle parameters (width over length) and converting it to nm units from pixel units.

**Traction force microscopy analysis.** Cell-generated traction forces were quantified using Image J 2.14.0 v (National Institute of Health, US) and several plug-ins available open-source, following the protocol developed by Qingzon Tseng[70,71]. First, images were loaded onto ImageJ and stacked per channel, resulting in three different stacks: beads pre cell detachment (before), beads post cell detachment (after) and brightfield cell images. A maximum z-projection was created of the area closest to the cell-bead interface and saved. Then, the before and after images were opened and aligned with the Template Matching plugin. After the images were aligned, the displacement of the beads between the before and after pictures was calculated with the Particle Image Velocimetry (PIV) plugin. From PIV, a data file of displacement field vectors is obtained, which can be transformed into traction forces with the Fourier Transform Traction Cytometry (FTTC) plugin. With the FTTC plugin, the pixel size was scaled to microns using the appropriate conversion. Furthermore, the Poisson ratio and Young's modulus of the hydrogels were specified per hydrogel in order to accurately calculate the forces exerted by the cells. Poisson ratio was always input as 0.5 and the Young's modulus of each gel family was input as 0.4 kPa (soft) and 25KPa (stiff). The average forces of each individual cell were summed (excluding those below 0.5 Pa) and plotted. Stress maps were created with the ParaView (v5.8.0) software, to visualise the force distribution in the cells.

**YAP nuclear localisation analysis.** The nuclear to cytoplasmic YAP ratio was determined following Eq. (1)

$$\frac{nucYAP}{cytoYAP} = \left(\frac{nucYAP}{nucA}\right) \Big/ \left(\frac{cytoYAP}{cytoA}\right) \tag{1}$$

Where nucYAP is the integrated density of the YAP channel in the nucleus, nucA, the area of the nucleus (obtained with the dapi channel); cytoYAP the integrated density of YAP in the cytoplasm, calculated as: cytoYAP = cellYAP − nucYAP (being cellYAP the integrated density of YAP in the cell as defined by the actin cytoskeleton channel. Similarly, cytoA is the area of the cell cytoplasm, defined as: cytoA = cellA − nucA, where cellA is the total cell area calculated from the actin cytoskeleton channel.

## Mitochondrial morphology analysis

The Mitochondrial Analyzer plugin was opened and the 2D threshold options were optimised according to the quality of the images obtained. The Thresholding was optimised and performed as follows: background was subtracted (rolling (microns) = 1); a sigma filter plus was applied to further reduce background noise and smooth object signal (radius = 0.5, 2.0 sigma); contrast was enhanced with 'enhance local contrast' (max slope = 2) and gamma was adjusted (value = 0.8) to correct remaining dim areas. The threshold method employed was weighted mean with a block size of 1.25 μm and the $C$ value was between 5 and 7 (this required adjustment from one image set to another, but it was empirically determined with the '2D threshold optimise' menu available in the plugin). Finally, the 2D threshold menu has a few post-processing command options, of these, remove outliers (radius = 0.5 pixels) and show comparison of threshold to original were ticked. After a binary image of the mitochondria was obtained, the 2D analysis menu was clicked. Here, the per-cell analysis was performed as well as the per-mito analysis.

## Computational model

Details of the modified molecular clutch model are provided in the supplementary data (Supplementary Note 1, Supplementary Table S1). To compare computational results to those obtained experimentally, integrin density model values were scaled to the experimental length values and plotted as FA length (μm), by scaling one data point (scRNA stiff V−) to fit experimental values. The same scaling (integrin density of 84.1 arb.units equated to 1 μm) was applied to all integrin density values.

## Statistics and reproducibility

Statistical analysis and graph plots were performed using GraphPad version 9.0.0 and R studio software. Unless stated otherwise, when two populations were contrasted, a $t$-test was performed. In the case of normal data distribution, unpaired $t$-test with Welch's correction was performed; if data was not normally distributed, a Mann-Whitney $t$-test was performed. Normal distribution was assessed with D'Agostino Pearson normality tests. To compare several groups i.e. data per hydrogel pair, a two-way ANOVA was performed with Tukey multiple comparison correction. Unless stated otherwise, each experiment comprised three biological replicates (N). $P$ values indicating significance, ns > 0.05, *≤0.05, **≤0.01, ***≤0.001, ****≤0.0001.

## Reporting summary

Further information on research design is available in the Nature Portfolio Reporting Summary linked to this article.

# Data availability

All data are available in the main text or supplementary materials. The transcriptomics raw and processed datasets from this work have been deposited to Gene Expression Omnibus with the accession codes GSE288423, GSE288464 and GSE288465. Source data are provided with this paper.

# Code availability

Computational simulation code available from the corresponding authors (P.R.-C.) upon request. Analysis scripts are available from the authors (M.A.G.O.) upon request.

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

## Acknowledgements

We thank the Gladstone Bioinformatics Core for their interactive enrichment analysis tools. M.A.G.O. acknowledges the Wellcome grant [204820/Z/16/Z], which supported the RNA sequencing shown in this work. S.D. acknowledges a Worldwide Cancer Research grant 21-0156, AIRC Foundation investigator grants 21392 and 28940, and Italian Ministry of University and Research PRIN grants 2022T9RM8A and P2022CE7SP. P.R. acknowledges 2020, 2021 and 2022 Veronesi Foundation Postdoctoral Fellowships and an AIRC MFAG 27453. We acknowledge financial support from the European Research Council AdG (Devise, 101054728 to M.S.S) and EPSRC HT2050 grant (EP/X033554/1 to M.S.S.), Spanish Ministry of Science and Innovation (PID2022-142672NB-I00 to P.R.-C. and MCIN/AEI/10.13039/501100011033 through the PID2022-136433OB-C21 to M.S.-S.), the European Research Council (grant 101097753 MechanoSynth to P.R.-C.), the Generalitat de Catalunya (2021 SGR 01425 to P.R.-C.), The prise 'ICREA Academia' for excellence in research to P.R.-C. IBEC is a recipient of a Severo Ochoa Award of Excellence from MINCIN and member of CERCA Programme/Generalitat de Catalunya.

## Author contributions

M.A.G.O., M.V., M.S.S. conceived the project. O.D., M.V. and M.S.S. supervised the project. M.A.G.O. performed all experiments and analysed all experimental data. P.R. and S.D. supervised the metabolic experiments performed in the University of Padova. G.C. and J.L.V. performed mechanical characterisation and data analysis together with M.A.G.O. G.F. and P.R.-C. led the development of the computational model and reported obtained data. J.L. developed and shared the protocol for the synthesis of soft V- and soft V+ matrices. P.G. provided the immortalised Y201 MSC cell line. M.A.G.O. wrote the first original draft of the article, which was reviewed and edited by G.C. and M.S.S. The article was read and corrected by all authors, who contributed to the interpretation of results. Funding was acquired by M.S.S., P.R.-C., S.D. and M.A.G.O.

## Competing interests

The authors declare no competing interests.
