## [Transparent Peer Review file · Nature Communications]

Piezo1 regulates the mechanotransduction of soft matrix viscoelasticity

Corresponding Author: Professor Manuel Salmeron-Sanchez

Version 0:

Reviewer comments:

Reviewer #1

(Remarks to the Author)

This interesting manuscript by Oliva et al., interrogates the role of the mechanosensitive channel PIEZO1 in sensing changes in matrix viscoelasticity. This is done by combining an immortalised mesenchymal stem cell line and some interesting bioengineering approaches. The data is well presented and clearly the experiments are rigorous and well controlled. My only major query comes from the use of the word "mechanosensor". All the data points to this protein being important for the cellular outputs and the authors do an excellent job of showing this and for this reason I believe the manuscript is of great interest to multiple fields. However as discussed below I think the title should be softened and the word "mechanosensor" replaced prior to publication.

1. "Mechanosensor of ..."

Of course, I am not questioning the role of PIEZO1 as a mechanosensor, it can be purified and reconstituted and still responds to force (Syeda et al., 2016 Cell reports) the very definition of a mechanosensor. However, there is little to no evidence that PIEZO1 changes activity directly in response to matrix stiffness or other properties of the matrix (Ca²⁺ imaging etc). Thus, I think the title should be softened to remove the word mechanosensor but still reflect the importance of PIEZO1 in the process. This is not semantics, it clearly reflects our understanding of the biological process and to this reviewer it is premature to use the strong current title!

2. RNAseq

The RNAseq really seems like a late addition and does not dovetail well with the rest of the manuscript. Is there any way to probe specific genes that come out of the RNAseq so that it can better fit the narrative rather than feeling like a stand-alone piece of information added for 'omics sake rather than for true biological insight.

Other points:

-Line 228 "However, within the viscoelastic conditions, increasing stiffness (soft V + versus stiff V + conditions) integrin levels decrease experimentally'

It may be present but I couldn't identify where the experimental evidence for this is in the manuscript?

-Line 428 "Our data thus indicate that mitochondria are the main sensor of matrix mechanics in terms of cellular respiration." The terminology here again is less than optimal. How are the mitochondria the sensors, if PIEZO1 is the sensor? Surely this means mitochondria are the main target of P1 signalling they can't be "sensors" within the authors narrative.

-Line 448 "But does not induce a state of oxidative stress."

Where is the data to support this statement?

Reviewer #2

(Remarks to the Author)

The authors used two pairs of viscoelastic polyacrylamide hydrogels with tunable stiffness and viscoelasticity, for investigating the roles of Piezo1 in responding to viscoelastic cues of matrix. They demonstrate that Piezo1 is a mechanosensor of viscoelasticity in soft ECMs using a modified viscoelastic molecular clutch model. Then, they also showed the differentially regulated genes, which mediate the cellular response to stiffness, viscoelasticity, and Piezo1 expression via RNA sequencing. Overall, this article emphasized the importance of Piezo1 activity in a soft viscoelastic matrix. However, there are several confusing questions, and some additional work is necessary before further consideration.

1. It's important that the authors established a hydrogel system that can independently modulate substrate stiffness and viscoelasticity. This provides a feasible research model to investigate the role of stiffness and viscoelasticity affecting on cell

performance. However, this stiffness and viscoelasticity in 2D model to affect cell spreading and differentiation has been previously reported. Furthermore, some models have been established in a 3D matrix with tunable stiffness and viscoelasticity. Based on the tunable stiffness and viscoelasticity design, the novelty of this material system is limited. Besides, although the authors demonstrated the role of Piezo1 as a mechanosensor of viscoelasticity, this concept has been proved in previous literatures. This study hasn't given an obvious new finding.

2. The authors demonstrated that Piezo1 is a mechanosensor of viscoelasticity in soft ECMs. Then how about the role of Piezo1 in stiff ECMs? Can any possible mechanism be demonstrated from this study how the Piezo1 in sensing viscoelasticity signals in stiff ECMs? The authors should give some discussion for this, as the authors have designed the stiff substrate in this study. If Piezo1 shows little role in this process from the experimental data, how about the underlined mechanism?

3. The authors claimed that they established a modified viscoelastic molecular clutch model, however, the details for describing the components of this modified clutch model are missing in the text. What's the key of this model, and how about the significant difference of this model from previous reported models, such as the "Motor-clutch" models? The important improvement or feasibility of this model should be discussed and compared with previous models.

4. Line 230-232, the authors claimed that "To explain this in the model, this requires considering that integrin recruitment upon talin unfolding is a time-dependent process as well, which is captured by the model via a stochastic, non-instantaneous, binding of vinculin to unfolded talin." Dose any evidences show the talin unfolding is a time-dependent process? And how the vinculin binds to the unfolded talin in this model? More compressive discussion should be given.

5. For Fig. 2b and 2c, the number of statistical samples is " $n \geq 17$ ". Are the 17 images, 17 parallel samples, or 17 cells? If there are only 17 cells used for statistical analysis, the statistical sample size seems insufficient.

6. For Fig.S4c and Fig.3c, there was a significant difference in the actin retrograde flow of siPiezo1 MSCs between stiff (V-) and stiff (V+) (Fig.3c), while there were no significant differences within the traction forces of these cells (traction forces)? Actin retrograde flow velocity exhibits a dynamic negative correlation with cell traction force. What are the possible reasons for this phenomenon?

7. Though the authors explain the RNA-seq analysis of siPiezo1 and scRNA Y201 MSCs culture on soft and stiff matrices with different viscoelasticity. They seem not explain the roles of mechanical transduction related pathways in different groups. I am also confusing why the authors discuss the immune response pathways of siPiezo1 and scRNA Y201 MSCs on soft matrices? What's the relationship between the Piezo1 mechanosensing and immune response pathways? Dose the immune response pathway influences the mechanical transduction of MSCs?

8. Line 203, "MSCs" should be "MSCs".

Reviewer #3

(Remarks to the Author)

In this super impressive study, Oliva, M.A.G. and colleagues make an important leap in our ability to model biophysical properties in controlled in vitro and in silico settings. The work establishes a new protocol for the generation of hydrogels with tunable elastic and viscous moduli, demonstrating a greater dissipation of energy when an additional linear polyacrylamide component was added into the widely used acrylamide/bis-acrylamide hydrogel mix. These new viscoelastic hydrogels were then compared to their elastic counterparts in a series of experiments that demonstrate an increased spreading of cells on the soft (0.4 kPa) viscoelastic hydrogels, compared to the elastic, while spreading was disrupted by knock-down of Piezo1 or increasing the stiffness for the viscoelastic hydrogels to 25 kPa. They then went on to adapt their previous computational model to include a time-dependent relaxation after an initial resistance to deformation, thus incorporating a viscoelastic component that they could then use to recapitulate their findings and assess the effect(s) of different perturbations on the simulations with this new component.

While the findings are promising, some concerns given below should be addressed prior to accepting the study for publication.

1. Does the addition of the linear Aam change the topography of the hydrogel, and would this be play a role in assisting cells to adhere on the "soft" V+ condition? Did you observe different heights, for example, when you were performing the nanoindentation between adjacent regions?

2. Related to the modelling, it is unclear how many simulations were run for each parameter assessed. Is it that the model always gives the same value, or is it possible to run some additional simulations to give an idea of the variability and how this relates to the errors that are given with the in vitro data?

3. The use of a single siRNA for Piezo1 throughout the manuscript raises the concern that these effects may be caused by an off-target, especially given that the knock-down is only resulting in an ~50% reduction in protein and mRNA levels. Has this siRNA been used with another in a previous manuscript? If not, it would be beneficial to see a comparison of the knock-down efficiency of an additional siRNA against Piezo1 at the protein/mRNA level, and a validation that the effect is the same for some experiments, such as on the cell area and focal adhesion count presented in Supp Data Fig. 2. This would provide some peace-of-mind for the reader that these effects are indeed specific to Piezo1.

4. For the quantification of mitochondrial area per cell in Supp. Data Fig. 8, the effect of cell area is likely playing a large role in the assessment. Given that the cells are rounded up, is it instead possible to compare the signal from the mitochondria between the different conditions, or to capture the whole-cell volume to assess the change in mitochondrial area also in Z? This would support the claim that the differences in metabolic activity are due to a change in the area of mitochondria between the samples, and not just changes in the functionality of those that are present.

5. The use of very small cell numbers in some cases is a concern, particularly where the number of biological replicates is not given. For instance, the in cell western in Supp. Data Fig. 2b give the number of replicates clearly as "N=2, n≥10,000 cells". However, in microscopy experiments the number of cells is all that is given. For example, in Supp. Data Fig. 2f, "n≥48". It would be necessary to indicate the number of biological replicates that these cells came from and in several cases increase the number of cells that were imaged. In particular, in Supp. Data Fig. 3 the cell number assessed is as low as 7 and it is unclear whether this is due to the challenges of producing the gels or some other limiting factor, but if not, the number should be increased to improve the confidence in the data presented. While 10k cells would be excessive for these experiments, a minimum of 10 cells/biological replicate seems feasible from the experimental approaches and would make 30 cells a reasonable minimum for all experiments, particularly where 3 biological replicates are commonly expected for reliable comparisons to be made.

6. In the Materials and methods, it is unclear in the "PAAm hydrogel synthesis" section whether the final concentration of linear acrylamide in the hydrogel mix is 5%, or that the solution that was made yielded 5% linear acrylamide that was then mixed with the acrylamide/bis-acrylamide. Could you please clarify the modification to the protocol where linear acrylamide is added to the acrylamide/bis-acrylamide mix.

Version 1:

Reviewer comments:

Reviewer #1

(Remarks to the Author)

The authors have addressed all my queries, and I now support publication.

Reviewer #2

(Remarks to the Author)

I greatly appreciate the reviewer's constructive and insightful feedback. I am very satisfied with the responses and the outcome of the review process. Based on that, I sincerely hope that the Editorial Board will consider accepting this manuscript for publication.

Reviewer #3

(Remarks to the Author)

The authors have addressed all of my previous concerns.

REVIEWER COMMENTS – authors' response

We thank the reviewers and the Editor for their valuable and thoughtful comments, overall positive assessment and constructive feedback to improve the paper, which we have incorporated into our revised manuscript.

In the following, we individually address each of the reviewers' comments. We added lettering i.e. A, B and C, to aid in cross-referencing our responses. Responses to reviewers are in light blue, and modifications of the text in the revised manuscript will be highlighted in yellow.

A. Reviewer #1 (Remarks to the Author):

A. This interesting manuscript by Oliva et al., interrogates the role of the mechanosensitive channel PIEZO1 in sensing changes in matrix viscoelasticity. This is done by combining an immortalised mesenchymal stem cell line and some interesting bioengineering approaches. The data is well presented and clearly the experiments are rigorous and well controlled. My only major query comes from the use of the word "mechanosensor". All the data points to this protein being important for the cellular outputs and the authors do an excellent job of showing this and for this reason I believe the manuscript is of great interest to multiple fields. However as discussed below I think the title should be softened and the word "mechanosensor" replaced prior to publication.

A. We thank Reviewer #1 for taking the time to review our manuscript and we sincerely appreciate their recognition of the rigor of our work, the presentation of our data, and the acknowledgement of the potential impact of our research in multiple fields. Additionally, we appreciate their constructive feedback regarding the need to soften our title, and this, plus the remaining comments will be accordingly addressed point-by-point below:

A1. "Mechanosensor of ..."

Of course, I am not questioning the role of PIEZO1 as a mechanosensor, it can be purified and reconstituted and still responds to force (Syeda et al., 2016 Cell reports) the very definition of a mechanosensor. However, there is little to no evidence that PIEZO1 changes activity directly in response to matrix stiffness or other properties of the matrix (Ca²⁺ imaging etc). Thus, I think the title should be softened to remove the word mechanosensor but still reflect the importance of PIEZO1 in the process. This is not semantics, it clearly reflects our understanding of the biological process and to this reviewer it is premature to use the strong current title!

A1. We thank Reviewer #1 for highlighting that while the title alludes to the role of Piezo1 as a cell mechanosensor, whether it actually senses and activates in response to matrix viscoelasticity has not been directly addressed in our work, which in turn is more focused on the channel's role as a key mediator for the cell to respond to matrix viscoelasticity, and in particular, to soft viscoelastic matrices. We are therefore fully in agreement with Reviewer #1 and thank them for this relevant observation. Thus, the title has been

changed to "Piezo1 regulates the mechanotransduction of soft matrix viscoelasticity", as we believe it better reflects our observations, while still being a short and concise title.

A2. RNAseq

The RNAseq really seems like a late addition and does not dovetail well with the rest of the manuscript. Is there any way to probe specific genes that come out of the RNAseq so that it can better fit the narrative rather than feeling like a stand-alone piece of information added for 'omics sake rather than for true biological insight.

A2. We thank reviewer #1 for this comment. We performed a more specific differential gene expression analysis in order to obtain gene-family specific gene expression information that would match previously presented data in the manuscript. Specifically, we built gene families relevant to data shown in the text and plotted the rlog-normalized expression per stiffness group. Heatmap data was compared and discussed with data shown in the manuscript to increase the cohesion of the text. Therefore, we modified **Fig 5 a** and **Fig 6 a** of the main manuscript and included the general heatmap with differentially expressed genes ($p < 0.05$) in the supplementary data (**Supplementary Data Fig. 10 and Fig. 11**), as we believe this still conveys information on inner-stiffness group transcriptome variations. We have also added to our results and discussion section to report gene family heatmap data. Discussion of the stiff group is in lines 508-522:

"Curated family heatmaps display z-score rlog-normalised expression for genes from five functionally defined relevant gene families: *integrins*, *actin*, *YAP*, *mitochondria* and *contractility*, which were used to create a visual representation of inner stiffness-group transcriptome variation. From this data it is possible to discern the strong interplay between Piezo1, integrins, actin cytoskeleton, and myosin-based contractility mechanisms in the cell at this stiffness range (25 kPa); as siPiezo1 generally decrease gene expression within these functional families (**Fig 5, a**). Also, in accordance with our data, integrin, actin and contractility subfamilies are generally overexpressed in V- conditions, and levels decrease in response to enhanced energy dissipation (V+). Additionally, we assessed mitochondrial and YAP functional groups. The YAP subfamily is overexpressed in scRNA V-, with both Piezo1 knock down and energy dissipation having a downregulatory effect, as shown in **Fig 4, g**. Further, we observed that Piezo1 knock down seems to produce a decrease in mitochondrial genes, as demonstrated experimentally in **Fig 4, k**. Finally, a general downregulation of functional gene families is observed in response to enhanced energy dissipation at this stiffness."

and discussion of the soft group is between lines 579-595:

"We conducted the same analysis for scRNA vs siPiezo1 MSCs cultured on soft substrates, and identified 177 DE genes (**Supplementary Data Fig. 11**), which suggests that when MSCs are cultured on soft substrates both Piezo1 expression and substrate energy dissipation produce a less pronounced effect on the transcriptome than when MSCs are cultured on stiff substrates. A curated heatmap of rlog normalised gene expression across relevant gene families was plotted (**Fig. 6, a**). Here, in the scRNA V+ conditions genes such as vinculin, integrin $\beta 1$ and $\beta 3$, Myosin11 and Rock2 show

overexpression; highlighting their role in mediating the mechanoactivation we observed experimentally in response to enhanced energy dissipation at this stiffness. Accordingly, these all became downregulated in siPiezo1 cells. Additionally, increased energy dissipation on scRNA MSCs upregulated YAP1 and TEAD associated genes, commensurate with data shown in **Fig 4, d**, where we show that although YAP does not become nuclear at this stiffness in response to increased energy dissipation, there is an increase in mechanotransductive mechanisms. Finally, in our mitochondrial gene family, two mitochondrial biogenesis genes (TFAM and NDUFS1) were overexpressed in scRNA V+ conditions, indicating that enhanced energy dissipation at this stiffness potentially increases the metabolic capacity of cells without inducing mitochondrial membrane damage; this is in line with data we showed in **Fig 4, j.** and **Supplementary Fig. 8.**”

The modified figures are now as follows:

Fig. 5. RNA-seq analysis of siPiezo1 and scRNA Y201 MSCs cultured on stiff elastic (V-) and viscoelastic (V+) matrices. (a) Curated z-score (rlog-

normalised expression) heatmap of stiff group genes across experimental conditions per defined gene families (Integrins, Actin cytoskeleton, YAP signaling, Mitochondrial and Contractility). **(b)** (Top) Heatmap of enriched results from Over Representation Analysis (ORA), which overlap with the most DE genes in the scRNA MSC V- vs scRNA MSC V+ comparison. (Bottom) A network of the three top enriched DE gene groups connected by their overlapping genes. **(c)** Heatmap of enriched results from ORA that overlap with the most DE genes in the siPiezo1 V- vs siPiezo1 V+ comparison. **(d)** Heatmap of enriched results from ORA that overlap with the most DE genes in the scRNA V- vs siPiezo1 V- comparison.

a SOFT

b

0.4 kPa scRNA V- vs scRNA V+

Family
 Integrins
 Actin
 YAP
 Mitochondria
 Contractility

Z-score
 2
 1
 0
 -1
 -2

c

0.4 kPa siPiezo1 V- vs siPiezo1 V+

d

0.4 kPa scRNA V- vs siPiezo1 V-

Fig.6. RNA-seq analysis of siPiezo1 and scRNA Y201 MSCs cultured soft elastic (V-) and viscoelastic (V+) matrices. (a) Curated z-score (log-

normalised expression) heatmap of soft group genes across experimental conditions per defined gene families (Integrins, Actin cytoskeleton, YAP signaling, Mitochondrial and Contractility). (b) (Top) Heatmap of enriched results from Over Representation Analysis (ORA), which overlap with the most DE genes in the scRNA MSC V- vs scRNA MSC V+ comparison. (Bottom) A network of the top enriched DE gene groups connected by their overlapping genes. (c) Heatmap of enriched results from ORA that overlap with the most DE genes in the siPiezo1 V- vs siPiezo1 V+ comparison. (d) Heatmap of enriched results from ORA that overlap with the most DE genes in the scRNA V- vs siPiezo1 V- comparison.

Other points:

A3. -Line 228 “However, within the viscoelastic conditions, increasing stiffness (soft V + versus stiff V + conditions) integrin levels decrease experimentally’

It may be present but I couldn’t identify where the experimental evidence for this is in the manuscript?

A3. We thank Reviewer #1 for this comment; indeed, we have not directly assessed integrin levels experimentally. Here we meant to refer to integrin recruitment to adhesions, which is measured experimentally through vinculin focal adhesion length and overall size (Fig. 2, Supplementary Data Fig. 3.). Still, the text should reflect this, and we have now changed the line from ‘However, within the viscoelastic conditions, increasing stiffness (soft V + versus stiff V + conditions) integrin levels decrease experimentally’ to line 243-245 ‘However, within the viscoelastic conditions, increasing stiffness (soft V+ versus stiff V+ conditions) reduces adhesion length (Fig. 2, d), and thus integrin recruitment.’

A4. -Line 428“Our data thus indicate that mitochondria are the main sensor of matrix mechanics in terms of cellular respiration.”

The terminology here again is less than optimal. How are the mitochondria the sensors, if PIEZO1 is the sensor? Surely this means mitochondria are the main target of P1 signalling they can’t be “sensors” within the authors narrative.

A4. We thank Reviewer #1 for this comment and for the helpful suggestion. It is true that mitochondria respond to changes in substrate mechanics but indeed they are not the sensors of these properties. Additionally, we wanted to highlight the role of the Piezo1 channel abrogation on mitochondrial respiration. We have therefore edited the text to better reflect this and maintain accurate terminology, in line with the definitions of mechanosensor used throughout the text and in existing literature. Lines 452-543: “Our data thus indicate that mitochondria are the main sensor of matrix mechanics in terms of

cellular respiration.” now reads: “Our data thus indicate that mitochondria are targeted by the Piezo1-modulated mechanosensing of matrix mechanics, regulating cellular respiration.” as proposed.

A5-Line 448 “But does not induce a state of oxidative stress.”
Where is the data to support this statement?

A5. We thank Reviewer #1 for this observant comment. In the text we discuss the effect of stress relaxation in stiff substrates on control scRNA MSCs. We observe that mitochondria size decreases; this is mainly since overall cell size decreases as well as mechanoactivation, in response to increased stress relaxation, at this stiffness. Still, we do not see significant differences between mitochondrial form factor (**Fig. 4 k. Stiff V- scRNA vs Stiff V+ scRNA**). We described in the text that mitochondrial elongation provides information on fusion vs fission events and thus alludes to the respiratory state or proliferative capability of cells (Chen et al., 2023), and this is what we wanted to convey with our statement ‘...but does not induce a state of oxidative stress’, meaning that mitochondrial mass is smaller and cells are not mechanoactive, but still maintain metabolic capabilities as their mitochondria are not fragmented. However, we agree that a statement such as “...but does not induce a state of oxidative stress” is perhaps too strong for the data we present, and we have modified the text to reflect our observations accordingly. Therefore, Line 448 “But does not induce a state of oxidative stress.” is now Lines 471-474: “...whereas on stiff matrices (**Supplementary Data Fig. 8, b**), increased stress relaxation decreases overall cellular respiration and slows the cell’s metabolic capabilities. Still, it does not significantly reduce mitochondrial form factor (**Fig. 4, k**), suggesting that cells are not under oxidative stress.”

B. Reviewer #2 (Remarks to the Author):

The authors used two pairs of viscoelastic polyacrylamide hydrogels with tunable stiffness and viscoelasticity, for investigating the roles of Piezo1 in responding to viscoelastic cues of matrix. They demonstrate that Piezo1 is a mechanosensor of viscoelasticity in soft ECMs using a modified viscoelastic molecular clutch model. Then, they also showed the differentially regulated genes, which mediate the cellular response to stiffness, viscoelasticity, and Piezo1 expression via RNA sequencing. Overall, this article emphasized the importance of Piezo1 activity in a soft viscoelastic matrix. However, there are several confusing questions, and some additional work is necessary before further consideration.

B. We thank Reviewer #2 for the time to review our work and provide insightful feedback. We will address each of their comments below:

B1. It’s important that the authors established a hydrogel system that can independently modulate substrate stiffness and viscoelasticity. This provides a feasible research model to investigate the role of stiffness and viscoelasticity affecting on cell performance. However, this stiffness and viscoelasticity in 2D model to affect cell spreading and

differentiation has been previously reported. Furthermore, some models have been established in a 3D matrix with tunable stiffness and viscoelasticity. Based on the tunable stiffness and viscoelasticity design, the novelty of this material system is limited. Besides, although the authors demonstrated the role of Piezo1 as a mechanosensor of viscoelasticity, this concept has been proved in previous literatures. This study hasn't given an obvious new finding.

B1. We agree with the reviewer that 2D hydrogel systems where *stiffness/elasticity and energy dissipation/viscoelasticity* can be independently modulated have been reported, some of the works include polyacrylamide systems such as this work from our lab (Walker et al., 2023) and other papers from the Janmey group (Charrier et al., 2020, 2018), who established the use of linear Aam in polyacrylamide to modulate dissipative properties. We therefore do not claim novelty in the development of these substrates.

To respond to the reviewer comment in relation to the new finding of this study, no previous work has addressed the role that piezo1 might play in relaying physiological viscoelastic matrix cues, and most importantly, there have been no previous computational models of the molecular clutch that coupled the channel's concomitant action with integrins in the actin-talin-integrin-fibronectin clutch. The novelty of our work lies within the application of viscoelastic hydrogels to systematically dissect cell response within the molecular clutch framework in a soft vs stiff context; as well as the inclusion of the Piezo1 within this model, both experimentally and computationally. The paper thus reveals the role of Piezo1 in this process of mechanotransduction in soft viscoelastic matrices.

B2. The authors demonstrated that Piezo1 is a mechanosensor of viscoelasticity in soft ECMs. Then how about the role of Piezo1 in stiff ECMs? Can any possible mechanism be demonstrated from this study how the Piezo1 in sensing viscoelasticity signals in stiff ECMs? The authors should give some discussion for this, as the authors have designed the stiff substrate in this study. If Piezo1 shows little role in this process from the experimental data, how about the underlined mechanism?

B2. We thank Reviewer #2 for this comment. This is a valid consideration; indeed, our paper is focused on the important role Piezo1 plays in cell's response to a viscoelastic soft matrix. In our manuscript, we touch on how Piezo1 generally suppresses cell-substrate interaction on stiffer substrates as well as on FN coated glass controls, see Lines 186-194: "*In this study, siPiezo1 MSCs decreased their spreading area and increased their circularity, as compared to scRNA MSCs, when cultured on soft viscoelastic (V+) or stiff elastic (V-) hydrogels (Fig. 1, g, h). Interestingly, the cell-spreading response to viscoelasticity seen in scRNA MSCs was abrogated in siPiezo1 MSCs cultured on low stiffness (soft, 0.4 kPa) but not on stiff substrates (stiff, 25 kPa), as these cells had presumably reached their minimum spreading area independently of Piezo1. This Piezo1-mediated cell spreading behaviour was also observed when we cultured MSCs on FN-coated glass substrates (Supplementary Data, Fig. 2, d, e and f), where siPiezo1 cells showed reduced cell spreading area compared to scRNA MSCs.*

Considering this, we believe that there is no difference between our siPiezo1 and scRNA MSCs when cultured on stiff V+ substrates due to the cell never being able to reach a *mechano-active* state, which we observed experimentally and from computational models that predicted molecular clutch activation (**Fig 2** and **Fig 3**). Because of this, Piezo1 cannot further potentiate cell-substrate interaction by enhancing adhesion complex recycling, as integrins are not bound. Still, we agree that although this is assumed by us and discussed more extensively in recent works such as (Huerta-López et al., 2024), it warrants additional discussion in the text, and that while we chose to focus on soft viscoelastic matrices, we must address our data sets fully. We have therefore added the following to the manuscript:

Lines 671-673: "Faster stress relaxation on stiff substrates prevented mechanoactivation – reported also recently by other groups⁶¹. We therefore believe that Piezo1 is not able to influence cell response in a stiff viscoelastic regime, as it cannot potentiate adhesion complexes if integrins are not first recruited."

B3. The authors claimed that they established a modified viscoelastic molecular clutch model, however, the details for describing the components of this modified clutch model are missing in the text. What's the key of this model, and how about the significant difference of this model from previous reported models, such as the "Motor-clutch" models? The important improvement or feasibility of this model should be discussed and compared with previous models.

The model introduced in this work is indeed an implementation of the motor-clutch model. The model is based on our own previous work, where we developed a motor-clutch model to understand how cells sense the stiffness of their substrate, in a purely elastic context (Andreu et al., 2021; Elosegui-Artola et al., 2016; Oria et al., 2017). There are three main modifications with respect to our previous work. First, the substrate is modelled as a viscoelastic standard linear solid, instead of an elastic solid as done previously. Second, the timescale introduced by viscoelasticity necessitates accounting for the temporal dynamics of integrin recruitment as well, which we address by reducing the vinculin binding time, which we previously assumed to be instantaneous in our earlier work. Third, we introduce the role of piezo, which was not considered in our previous work. This is done simply by assuming that piezo activates integrins (as supported by previous work) by decreasing integrin unbinding rates (k_{off}). The model is introduced in the main text (lines 217-261) and then described in detail in the supplement (Supplementary Note 1). However, we have realized that we did not clarify in the main text what the additions were with respect to our previous model. We apologize for this, we have now clarified this in lines 232-236:

"To model the response to a viscoelastic substrate, we modified our previous model to consider a viscoelastic rather than elastic substrate, modelled as a Standard Linear Solid (SLS). The SLS exhibits instantaneous resistance to deformation and time-dependent relaxation, as observed experimentally (see **Fig. 1, c, d**), for details see **Supplementary Note 1.**"

B4. Line 230-232, the authors claimed that “To explain this in the model, this requires considering that integrin recruitment upon talin unfolding is a time-dependent process as well, which is captured by the model via a stochastic, non-instantaneous, binding of vinculin to unfolded talin.” Dose any evidences show the talin unfolding is a time-dependent process? And how the vinculin binds to the unfolded talin in this model? More compressive discussion should be given.

We apologize for not explaining this clearly. All the molecular events involved in clutch engagement are time-dependent in the sense that they occur at a given timescale, characterized by the corresponding binding, unbinding, and unfolding rates. In our previous models, we explicitly considered the binding and unbinding rates of integrins, and the unfolding rate of talin, all taken from experimental values from the literature. The last step in the engagement of the clutch is the binding of vinculin to unfolded talin, which is then assumed to lead to integrin recruitment. This is a step in our model that encompasses several molecular events, which include vinculin binding to talin and the downstream signalling that leads to integrin recruitment. In our previous work on elastic substrates, we had assumed for simplicity that this step was instantaneous. In the context of purely elastic substrates, in which forces do not dissipate due to viscous relaxation, this was sufficient to capture experimental behaviour. However, this step, like all others in the model, has a corresponding timescale. In the context of viscoelasticity of this work this timescale is relevant, as it may compete with the timescale of force relaxation. Consequently, the best fit of the model was obtained by implementing a more realistic non-instantaneous binding rate between vinculin and talin. We have now clarified this in the text in lines 243-249:

‘To explain this in the model, this requires considering that integrin recruitment does not occur instantaneously upon talin unfolding, as assumed for simplicity in our previous models on elastic substrates. Instead, here we explicitly consider the timescale of this process, captured by the model via a stochastic, binding rate of vinculin to unfolded talin.’

B5. For Fig. 2b and 2c, the number of statistical samples is “ $n \geq 17$ ”. Are the 17 images, 17 parallel samples, or 17 cells? If there are only 17 cells used for statistical analysis, the statistical sample size seems insufficient.

B5. We thank Reviewer #2 for this comment. To ensure that our data is indeed representative, we increased our statistical sample sizes where possible. We have also clarified sample number (n) and experiment number (N) in all figure captions.

B6. For Fig.S4c and Fig.3c, there was a significant difference in the actin retrograde flow of siPiezo1 MSCs between stiff (V-) and stiff (V+) (Fig.3c), while there were no significant differences within the traction forces of these cells (traction forces)? Actin retrograde flow velocity exhibits a dynamic negative correlation with cell traction force. What are the possible reasons for this phenomenon?

B6. We thank Reviewer #2 for this comment. It is true that actin retrograde flow velocity should be inversely correlated to cell exerted forces (tractions), however, as we have addressed in the text, we have not used a viscoelastic algorithm to calculate cell-exerted tractions on our V+ substrates, which likely overestimates the forces that cells are able to produce on these matrices. We do report that siPiezo1 cells on **stiff V-** substrates exert traction forces of approximately 5Pa, whereas on **stiff V+** substrates this seems to increase to >10Pa (**Supplementary Data Fig. 4. d**), still, within the stiff condition, this is not a statistically significant change. Previous work has touched on the importance of Piezo1-mediated Ca²⁺ signaling in mediating traction force generation, as a sort of feedforward loop between traction forces eliciting Piezo1 channel activation, and in turn, having Piezo1 mediated Ca²⁺ signaling enhancing the activity of myosin light chain kinase (MLCK), further enhancing myosin II contractility (Ellefsen et al., 2019). Thus, it is possible that Piezo1 channel KD blunts traction force generation to similar levels in stiff V- and stiff V+ matrices when these forces are overestimated in V+ conditions. Still, when comparing **soft V-** vs **stiff V-** traction force generation, we do observe a substantial increase in cell-generated forces (**Supplementary Data Fig 4. d**) and this is reflected also in a decrease of flow speed (**Fig 3, d**).

B7. Though the authors explain the RNA-seq analysis of siPiezo1 and scRNA Y201 MSCs culture on soft and stiff matrices with different viscoelasticity. They seem not explain the roles of mechanical transduction related pathways in different groups. I am also confusing why the authors discuss the immune response pathways of siPiezo1 and scRNA Y201 MSCs on soft matrices? What's the relationship between the Piezo1 mechanosensing and immune response pathways? Dose the immune response pathway influences the mechanical transduction of MSCs?

B7. We thank reviewer #2 for this comment. Please see comment A2 regarding more specific mechanical transduction related pathways being addressed in our RNAseq analysis.

Regarding the second part of the comment on immune pathways: our gene ontology (GO) analysis was meant to highlight the top differentially expressed gene sets between our conditions, to have an unbiased perspective on overall transcriptome changes between our experimental conditions. Indeed, we observed that between our scRNA and siPiezo1 cells on soft V+ matrices, the main differentially expressed gene modules were related to immune processes, which were downregulated in siPiezo1 cells, when compared to scRNA MSCs. As discussed, we found that previous work that had a similar transcriptomic approach to ours when comparing the transcriptome of MSCs encapsulated in 3D alginate hydrogels of $E \sim 3k Pa$ vs $E \sim 18 kPa$, where authors report a stiffness-induced activation of immune response via NF- κ B. Previous work from our lab has also described how intermediate adhesion phenotypes in MSCs are capable of inducing immunomodulatory effects, where cells are in a growing, self-renewal state (Dalby et al., 2018). We thus believe Piezo1 is closely involved in regulating this intermediate adhesion phenotype that MSCs reach when cultured on soft V+ substrates.

We will expand the discussion to include the following:

Lines 649-653: "Previous work from our group ^{56,61} has addressed that MSCs reach an immunomodulatory phenotype in *intermediate adhesion* states, where adhesion – and thus- molecular clutch mechanisms are active but not so overly engaged to promote differentiation; from these results we propose that Piezo1 is a mediator of stiffness-induced immunomodulation in MSCs on softer substrates."

B8. Line 203, "MSCs" should be "MSCs".

B8. We thank Reviewer #2 for this comment; we have now fixed this typo in the text, in line 236.

C. Reviewer #3 (Remarks to the Author):

C. In this super impressive study, Oliva, M.A.G. and colleagues make an important leap in our ability to model biophysical properties in controlled in vitro and in silico settings. The work establishes a new protocol for the generation of hydrogels with tunable elastic and viscous moduli, demonstrating a greater dissipation of energy when an additional linear polyacrylamide component was added into the widely used acrylamide/bis-acrylamide hydrogel mix. These new viscoelastic hydrogels were then compared to their elastic counterparts in a series of experiments that demonstrate an increased spreading of cells on the soft (0.4 kPa) viscoelastic hydrogels, compared to the elastic, while spreading was disrupted by knock-down of Piezo1 or increasing the stiffness for the viscoelastic hydrogels to 25 kPa. They then went on to adapt their previous computational model to include a time-dependent relaxation after an initial resistance to deformation, thus incorporating a viscoelastic component that they could then use to recapitulate their findings and assess the effect(s) of different perturbations on the simulations with this new component.

While the findings are promising, some concerns given below should be addressed prior to accepting the study for publication.

C. We thank Reviewer #3 for taking the time to review our manuscript and for their encouraging and constructive comments. We sincerely value the recognition of the potential impact of our work and will address each of their comments below:

C1. Does the addition of the linear Aam change the topography of the hydrogel, and would this be play a role in assisting cells to adhere on the "soft" V+ condition? Did you observe different heights, for example, when you were performing the nanoindentation between adjacent regions?

C1. We thank Reviewer #3 for this comment. We agree that this is a valid and necessary additional experiment, that would help strengthen our hydrogel characterization. We performed additional nanoindentation maps with the Chiaro system, which we also employed for characterizing the viscoelastic properties of our hydrogel system. By plotting the calculated contact point per each indentation curve, we get an approximation of the variations in topography of our gels. In this case, topographies were non significantly different between soft V- and soft V+, as well as stiff V- and stiff V+ substrates. This data has now been added to our supplementary file, please find attached updated **Supplementary Data Fig 1**:

With the updated caption:

Supplementary Data Fig. 1.

(a) Time for stress relaxation to occur to 80% of original stress value (s) plotted for Soft (grey) and Stiff (orange) hydrogel groups. ($n \geq 61$ curves, each dot represents a map of ≥ 4 single nanoindentation curves each) (b) G''/G' plot of bulk rheology frequency sweep measurements performed at 0.1% strain for the soft group. (c) G''/G' plot of bulk rheology frequency sweep measurements performed at 0.1% strain for the stiff group. Shown as mean \pm SD of $N = 3$ hydrogels. (d) Representative immunofluorescence images of the 10 $\mu\text{g/ml}$ fibronectin coating (red) of the hydrogels, showing the middle and edge of the hydrogel on the glass coverslip. Images of stained non-functionalised hydrogels are also shown in the empty column, for each hydrogel type. Scale bar 100 μm . (e) Quantification of the resulting fibronectin staining intensity (integrated density a.u.) shown as mean \pm SD of different areas of $n > 3$ of each hydrogel $N = 3$. (f) Topography measurements (μm) for soft (left) and stiff (right) hydrogel groups, shown as individual values $n \geq 385 \pm$ SD $N=3$ hydrogels of. (g) Representative 20x20 μm topography maps per hydrogel condition, Statistical analyses were performed using a two-way ANOVA (a), Kruskal-Wallis's test (e) and unpaired t-tests (f). P values indicating significance, ns > 0.05 , **** ≤ 0.0001 .

C2. Related to the modelling, it is unclear how many simulations were run for each parameter assessed. Is it that the model always gives the same value, or is it possible to run some additional simulations to give an idea of the variability and how this relates to the errors that are given with the in vitro data?

The simulations are implemented through a stochastic monte carlo algorithm. The stochastic nature of the algorithm means that each run gives a slightly different result. Following the reviewer suggestion, we have now updated this, running 26 simulations and including the SD of these results to our figures – See updated **Fig 2 (f-h) and Fig 3 (e-g)**.

C3. The use of a single siRNA for Piezo1 throughout the manuscript raises the concern that these effects may be caused by an off-target, especially given that the knock-down is only resulting in an $\sim 50\%$ reduction in protein and mRNA levels. Has this siRNA been used with another in a previous manuscript? If not, it would be beneficial to see a comparison of the knock-down efficiency of an additional siRNA against Piezo1 at the protein/mRNA level, and a validation that the effect is the same for some experiments, such as on the cell area and focal adhesion count presented in Supp Data Fig. 2. This would provide some peace-of-mind for the reader that these effects are indeed specific to Piezo1.

C3. We thank Reviewer #3 for this comment. When embarking on this project, we discussed how to best reduce Piezo1's influence on cells, starting with the mechanosensitive ion channel toxin GsMTx4 and eventually moving to siRNA, which we believed offered a more targeted inhibitory effect. This approach was similar to other key publications that touched on Piezo1 mechanosensing such as the work by the Pathak group (Pathak et al., 2014) or more recent works (Luo et al., 2022) that have used the same siRNA as us to target Piezo1 (# AM16708, Assay ID:138387, Thermo Fisher). Our data suggests that we were targeting the channel, as we saw clear phenotypic differences (**Supplementary Data Fig. 2, e.**), also reported by previous literature in response to Piezo1 silencing or knock down (McHugh et al., 2010). Furthermore, Piezo1 expression was significantly reduced both in terms of protein and gene level (**Supplementary Data Fig. 2, e, f**) Finally, when upscaling this transfection to perform RNA sequencing, we once again observed the down regulation of Piezo1 in our control dataset (**Supplementary Fig. 9., a, b.**), which gave us confidence to assume a reduction in Piezo1's influence in our siPiezo1 MSCs.

Still, to strengthen our data, we added experiments where we compare the level of channel knock down using different siRNAs from Thermo Fisher's library as well as the toxin GsMTx4. In addition to this, we also compared cell spreading area and focal adhesion formation on cells transfected with a control siRNA, and three different commercially available siRNAs (1, 2 and 3) as well as cells cultured with 10 μ M GsMTx4. We observed that all screened siRNAs reduced Piezo1 expression significantly (**Supplementary Data Fig. 2, a**). Additionally, all except siRNA 3, produced a significant reduction in cell area (**Supplementary Data Fig. 2, b**) and that all treatments reduced focal adhesion length, although in this case, siRNA 1 and siRNA 3 outperformed siRNA 2. From this data, we are confident that the siRNA used in our work (siRNA1) is capable of targeting Piezo1 in a specific and consistent way.

We have updated this discussion in the text in lines 182-192:

'We next investigated Piezo1 gene expression control Y201 MSCs (scRNA) and of MSCs in which Piezo1 was knocked down via small interfering RNA (siRNA) for Piezo1 using an siRNA screen (siRNAs 1-3) (**Supplementary Data, Fig. 2, a**). Additionally, we assessed morphology and adhesions on all screened conditions plus on cells treated with the mechanosensitive ion channel inhibitor GsMTx4 (**Supplementary Data, Fig. 2, b, c**). This allowed us to select the most consistent siRNA, siRNA1, which was thereafter subsequently used in all experiments and referred to as siPiezo1. siPiezo1 mediated Piezo1 knock down demonstrated clear phenotypic differences in cell morphology and adhesion formation (**Supplementary Data, Fig. 2, d**), as well as approximately 50% efficiency in Y201 MSCs in terms of protein and gene expression levels (**Supplementary Data, Fig. 2, e, f**).'

As well as a revised Supplementary Data Figure 2:

With the updated caption:

Supplementary Data Fig. 2.

(a) RT-qPCR of Piezo1 siRNA screen on Y201 MSCs. Data shown as mean fold change normalized to internal control RPL13A and to scRNA control, plotted as min to max box plot, showing all experimental data points. $n = 8$ technical replicates from $N = 4$ independent experiments. (b) Mean cell area from Piezo1 siRNA screen and GsMTx4 (10

μM) 48h culture post transfection. Data shown as min to max box & whiskers plot showing all individual points. $n \geq 18$ from $N = 2$ individual experiments. (b) FA length from Piezo1 siRNA screen and GsMTx4 (10 μM) 48h culture post transfection. Data shown as min to max box & whiskers plot showing all individual points. $n \geq 460$ adhesions from a minimum of 18 cells, from $N = 2$ individual experiments. (d) Representative images of siRNA1 (siPiezo1) and scRNA treated Y201 MSCs cultured on FN coated coverslips. Scale bar 50 μm (e) Piezo1 protein signal as quantified from in cell western, shown as individual experiment mean \pm SD, $N = 3$, $n \geq 10,000$ cells. (f) Mean fold change of Piezo1 gene expression normalised to RPL13A, shown as min to max box & whiskers, $n \geq 5$ from $N = 4$ experiments. Statistical analyses were performed using a one-way ANOVA. P values indicating significance, ns > 0.05 , * ≤ 0.05 , ** ≤ 0.01 , *** ≤ 0.001 , **** ≤ 0.0001 .

C4. For the quantification of mitochondrial area per cell in Supp. Data Fig. 8, the effect of cell area is likely playing a large role in the assessment. Given that the cells are rounded up, is it instead possible to compare the signal from the mitochondria between the different conditions, or to capture the whole-cell volume to assess the change in mitochondrial area also in Z? This would support the claim that the differences in metabolic activity are due to a change in the area of mitochondria between the samples, and not just changes in the functionality of those that are present.

C4. We thank Reviewer #3 for their comment. Cell size and mitochondrial content are inherently linked, as reported in previous literature (Kitami et al., 2012). Thus, with the data we show in **Supp Data Fig. 8**, our goal was to complement our data on mitochondrial function (assessed by mitochondrial form factor, reported in **Fig. 4, j-l**), with a proxy for mitochondrial content, using mitochondrial area per cell. These two parameters provide distinct but complementary information about cellular metabolism: i) mitochondrial content approximates the total respiratory capacity of the cell and ii) mitochondrial morphology (form factor/elongation) relates to mitochondrial function and dynamics, particularly the metabolic state and overall respiratory efficiency (Mishra and Chan, 2016). Together, these datasets allow us to hypothesize on expected changes in oxidative phosphorylation, akin to those measurable via a Seahorse oxygen consumption assay. Mitochondrial volume would provide another (even if more accurate) proxy for metabolic activity; still, we believe that it would not add significantly to the overall story. In order to further delve into mitochondrial function, we included a mitochondrial gene family in our RNAseq heatmap. This data supported metabolic outputs hypothesised from mitochondrial morphology data (see response to Reviewer 1, A2, **Fig 5a** and **Fig 6a**).

C5. The use of very small cell numbers in some cases is a concern, particularly where the number of biological replicates is not given. For instance, the in cell western in Supp. Data Fig. 2b give the number of replicates clearly as “ $N=2$, $n \geq 10,000$ cells”. However, in microscopy experiments the number of cells is all that is given. For example, in Supp. Data Fig. 2f, “ $n \geq 48$ ”. It would be necessary to indicate the number of biological replicates that these cells came from and in several cases increase the number of cells that were imaged. In particular, in Supp. Data Fig. 3 the cell number assessed is as low as 7 and it is unclear whether this is due to the challenges of producing the gels or some other

limiting factor, but if not, the number should be increased to improve the confidence in the data presented. While 10k cells would be excessive for these experiments, a minimum of 10 cells/biological replicate seems feasible from the experimental approaches and would make 30 cells a reasonable minimum for all experiments, particularly where 3 biological replicates are commonly expected for reliable comparisons to be made.

C5. We thank Reviewer #3 for this feedback. This comment is similar to Reviewer #2's, B5, which highlights the fact that we should indeed provide more precise information regarding cell and individual experiment numbers, in order to reinforce our claims. Plus, some experiments report few cell numbers. This is mainly due to the complexity of the imaging, as single-cell high resolution and sometimes live imaging is required for most of our experiments. Still, we have repeated the majority of our fixed cell imaging to strengthen our work. We have updated the following figures and their respective legends as suggested by Reviewer #3, as well as #2, specifying the number of cells and of biological repeats : Fig 1 g, h, i, j, k & l; Fig 2 b, c & d; Fig 4, a, b, d, e, f, g, h, j, k & l; Supplementary Data Fig 3, a, b & c.

C6. In the Materials and methods, it is unclear in the "PAAm hydrogel synthesis" section whether the final concentration of linear acrylamide in the hydrogel mix is 5%, or that the solution that was made yielded 5% linear acrylamide that was then mixed with the acrylamide/bis-acrylamide. Could you please clarify the modification to the protocol where linear acrylamide is added to the acrylamide/bis-acrylamide mix.

C6. We thank Reviewer #3 for this comment. We have now updated this section in our manuscript, to more accurately describe the preparation of the Linear Aam mix. See Lines 710-712: "making a final stock concentration of 5% linear acrylamide, which was then added to the soft V+ hydrogel solution, as described in **Supplementary Data Table S2.**"

Literature mentioned:

- Andreu, I., Falcones, B., Hurst, S., Chahare, N., Quiroga, X., Le Roux, A.-L., Kechagia, Z., Beedle, A.E.M., Elosegui-Artola, A., Trepas, X., Farré, R., Betz, T., Almendros, I., Roca-Cusachs, P., 2021. The force loading rate drives cell mechanosensing through both reinforcement and cytoskeletal softening. *Nat. Commun.* 12, 4229. <https://doi.org/10.1038/s41467-021-24383-3>
- Bavi, N., Richardson, J., Heu, C., Martinac, B., Poole, K., 2019. PIEZO1-Mediated Currents Are Modulated by Substrate Mechanics. *ACS Nano* 13, 13545–13559. <https://doi.org/10.1021/acsnano.9b07499>
- Charrier, E.E., Pogoda, K., Li, R., Park, C.Y., Fredberg, J.J., Janmey, P.A., 2020. A novel method to make viscoelastic polyacrylamide gels for cell culture and traction force microscopy. *APL Bioeng.* 4, 1–8. <https://doi.org/10.1063/5.0002750>

- Charrier, E.E., Pogoda, K., Wells, R.G., Janmey, P.A., 2018. Control of cell morphology and differentiation by substrates with independently tunable elasticity and viscous dissipation. *Nat. Commun.* 9, 449. <https://doi.org/10.1038/s41467-018-02906-9>
- Chaudhuri, O., Gu, L., Darnell, M., Klumpers, D., Bencherif, S.A., Weaver, J.C., Huebsch, N., Mooney, D.J., 2015. Substrate stress relaxation regulates cell spreading. *Nat. Commun.* 6, 6365. <https://doi.org/10.1038/ncomms7365>
- Chen, W., Zhao, H., Li, Y., 2023. Mitochondrial dynamics in health and disease: mechanisms and potential targets. *Signal Transduct. Target. Ther.* 8, 1–25. <https://doi.org/10.1038/s41392-023-01547-9>
- Dalby, M.J., García, A.J., Salmeron-Sanchez, M., 2018. Receptor control in mesenchymal stem cell engineering. *Nat. Rev. Mater.* 3, 1–14. <https://doi.org/10.1038/natrevmats.2017.91>
- Ellefsen, K.L., Holt, J.R., Chang, A.C., Nourse, J.L., Arulmoli, J., Mekhdjian, A.H., Abuwarda, H., Tombola, F., Flanagan, L.A., Dunn, A.R., Parker, I., Pathak, M.M., 2019. Myosin-II mediated traction forces evoke localized Piezo1-dependent Ca²⁺ flickers. *Commun. Biol.* 2, 298. <https://doi.org/10.1038/s42003-019-0514-3>
- Elosegui-Artola, A., Oria, R., Chen, Y., Kosmalska, A., Pérez-González, C., Castro, N., Zhu, C., Trepát, X., Roca-Cusachs, P., 2016. Mechanical regulation of a molecular clutch defines force transmission and transduction in response to matrix rigidity. *Nat. Cell Biol.* 18, 540–548. <https://doi.org/10.1038/ncb3336>
- Huerta-López, C., Clemente-Manteca, A., Velázquez-Carreras, D., Espinosa, F.M., Sanchez, J.G., Martínez-del-Pozo, Á., García-García, M., Martín-Colomo, S., Rodríguez-Blanco, A., Esteban-González, R., Martín-Zamora, F.M., Gutierrez-Rus, L.I., Garcia, R., Roca-Cusachs, P., Elosegui-Artola, A., del Pozo, M.A., Herrero-Galán, E., Sáez, P., Plaza, G.R., Alegre-Cebollada, J., 2024. Cell response to extracellular matrix viscous energy dissipation outweighs high-rigidity sensing. *Sci. Adv.* 10. <https://doi.org/10.1126/sciadv.adf9758>
- Kitami, T., Logan, D.J., Negri, J., Hasaka, T., Tolliday, N.J., Carpenter, A.E., Spiegelman, B.M., Mootha, V.K., 2012. A Chemical Screen Probing the Relationship between Mitochondrial Content and Cell Size. *PLOS ONE* 7, e33755. <https://doi.org/10.1371/journal.pone.0033755>
- Luo, M., Cai, G., Ho, K.K.Y., Wen, K., Tong, Z., Deng, L., Liu, A.P., 2022. Compression enhances invasive phenotype and matrix degradation of breast Cancer cells via Piezo1 activation. *BMC Mol. Cell Biol.* 23, 1. <https://doi.org/10.1186/s12860-021-00401-6>
- McHugh, B.J., Buttery, R., Lad, Y., Banks, S., Haslett, C., Sethi, T., 2010. Integrin activation by Fam38A uses a novel mechanism of R-Ras targeting to the endoplasmic reticulum. *J. Cell Sci.* 123, 51–61. <https://doi.org/10.1242/jcs.056424>
- Mishra, P., Chan, D.C., 2016. Metabolic regulation of mitochondrial dynamics. *J. Cell Biol.* 212, 379–387. <https://doi.org/10.1083/jcb.201511036>
- Oria, R., Wiegand, T., Escribano, J., Elosegui-Artola, A., Uriarte, J.J., Moreno-Pulido, C., Platzman, I., Delcanale, P., Albertazzi, L., Navajas, D., Trepát, X., García-Aznar, J.M., Cavalcanti-Adam, E.A., Roca-Cusachs, P., 2017. Force loading explains spatial sensing of ligands by cells. *Nature* 552, 219–224. <https://doi.org/10.1038/nature24662>

- Pathak, M.M., Nourse, J.L., Tran, T., Hwe, J., Arulmoli, J., Le, D.T.T., Bernardis, E., Flanagan, L.A., Tombola, F., 2014. Stretch-activated ion channel Piezo1 directs lineage choice in human neural stem cells. *Proc. Natl. Acad. Sci.* 111, 16148–16153. <https://doi.org/10.1073/pnas.1409802111>
- Walker, M., Pringle, E.W., Ciccone, G., Oliver-Cervelló, L., Tassieri, M., Gourdon, D., Cantini, M., 2023. Mind the Viscous Modulus: The Mechanotransductive Response to the Viscous Nature of Isoelastic Matrices Regulates Stem Cell Chondrogenesis. *Adv. Healthc. Mater.* <https://doi.org/10.1002/adhm.202302571>